# Explicit Planning Helps Language Models in Logical Reasoning

**Hongyu Zhao**[*1,3]   **Kangrui Wang**[1,3]   **Mo Yu**[2]   **Hongyuan Mei**[3]
[1]University of Chicago   [2]WeChat AI   [3]Toyota Technological Institute at Chicago
{hzhao,hongyuan}@ttic.edu

## Abstract

Language models have been shown to perform remarkably well on a wide range of natural language processing tasks. In this paper, we propose LEAP, a novel system that uses language models to perform multi-step logical reasoning and incorporates explicit planning into the inference procedure. Explicit planning enables the system to make more informed reasoning decisions at each step by looking ahead into their future effects. Moreover, we propose a training strategy that safeguards the planning process from being led astray by spurious features. Our full system significantly outperforms other competing methods on multiple standard datasets. When using small T5 models as its core selection and deduction components, our system performs competitively compared to GPT-3 despite having only about 1B parameters (i.e., 175 times smaller than GPT-3). When using GPT-3.5, it significantly outperforms chain-of-thought prompting on the challenging PrOntoQA dataset. We have conducted extensive empirical studies to demonstrate that explicit planning plays a crucial role in the system's performance.

## 1   Introduction

Logical reasoning is one of the most important and long-standing problems in artificial intelligence (Russell and Norvig, 2010). A logical reasoning system is able to draw new facts by applying known rules to known facts and determine the truth value of a given hypothesis; see Figure 1 for an example. For decades, research in building reasoning systems has heavily relied on formal logic. Since the surge of pretrained large language models (LMs), there have been efforts that harness the power of pretrained LMs and directly handle natural language statements to perform multi-step logical reasoning; see section 5 for a summary. In this paper, we propose LEAP, the first LM-based logical reasoning system that performs *explicit planning* during inference. While determining the truth value of a statement, our system searches over the known facts for those which

---

[*]Work done during internship at TTI-Chicago.

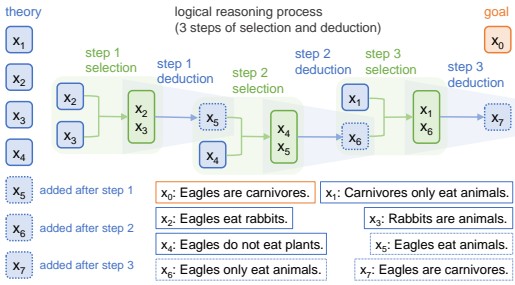

Figure 1: An example of theory $\mathcal{T}$ and goal $\mathbf{x}_0$ as well as a human-annotated multi-step logical reasoning process that proves the goal based on the theory.

are relevant and performs multiple rounds of deduction to reach the conclusion. At each round, the planning process looks ahead into the future outcomes of each possible reasoning decision (i.e., which to select and what to deduce), examining which of them is more likely to discover a valid proof for the given statement.

**Why planning?**   Planning is a fundamental property of intelligent behavior: it uses foresight to anticipate future outcomes of each possible decision and informs the process of decision making to achieve desirable end results. This concept has influenced the development of various methods in the field of artificial intelligence. Minimax-style game playing evaluates each possible move by anticipating replies and counterreplies between the player and the opponent (while assuming that both play optimally) (Russell and Norvig, 2010). Model-based reinforcement learning uses environment models to simulate responses to actions and then uses the simulated experiences to help learn value functions (e.g., Dyna, Monte-Carlo tree search) (Sutton and Barto, 2018). In natural language processing, planning has been used to help language models generate utterances that satisfy complex constraints (Lu et al., 2022a).

Planning is important for logical reasoning. By examining the future outcomes of each possible decision, a planning-based system will be able to focus on the actually useful (given and deduced) facts at early steps, thus enjoying a high chance of success. In addition, a planning-based reasoning system tends to be more interpretable, thus more useful in user-centric and safety-critical scenarios. For example, at each round of deduction, planning will explicitly show "what will happen after—and that is also why—I select these known facts

and deduce this particular new fact from them", which is more informative than only saying "I select these and deduce this." However, none of the previous LM-based systems use explicit planning during inference.

**Why is it challenging?** During planning, a verification mechanism is in need to determine the quality of each possible proof. In reality, the verification has to be performed by a model (like in model-based reinforcement learning), and models are imperfect due to architectural biases and finite training data. As a consequence, the reasoning system faces the problem of model exploitation: any model mistake may misguide the planning such that it favors a seemingly promising decision over the actually correct one. For example, the model may incorrectly think a statement proves the hypothesis, just because of a significant lexical overlap, causing the planning to favor a decision that helps deduce that statement and lead to the wrong conclusion.

**Our contributions.** We first propose a logical reasoning system along with a beam-search-style inference algorithm (section 3.1): the system utilizes pretrained LMs and mimics human-like step-by-step reasoning. Then we integrate explicit planning into the inference algorithm (section 3.2) and significantly improve the performance of the system. We empirically demonstrate that planning encounters the issue of model exploitation: when the given hypothesis is false, planning may find out an incorrect proof that fools the system to believe that the hypothesis is true. Finally, we develop a training strategy that effectively mitigates the issue of model exploitation (section 3.3). Our training strategy is adversarial: for each training theory, we synthesize a non-provable hypothesis but call the planning-based inference method to find a highly-scored proof for it; then we refine the verification model such that the score it assigns to that proof is suppressed; at the same time, we force the verification model to preserve its scores on the correct proofs of the provable hypothesises. Our experiments show that this strategy further significantly improves the performance of our system.

## 2 Problem Formulation

We consider the problem of logical reasoning. Given a hypothesis (or, in other words, a goal) $\mathbf{x}_0$ and a theory $\mathcal{T} = \{\mathbf{x}_1, \dots, \mathbf{x}_N\}$, we are interested in determining the truth value of $\mathbf{x}_0$, i.e., whether $\mathbf{x}_0$ can be logically proved by $\mathcal{T}$. If the goal $\mathbf{x}_0$ is provable, we are interested in discovering the reasoning process that proves it. Below is an example theory $\mathcal{T}$

$$\left\{ \begin{array}{ll} \text{``Richard is a King.''} & \text{``John is also a King.''} \\ \text{``John is greedy.''} & \text{``A greedy King is evil.''} \end{array} \right\}$$

For the goal "John is evil.", humans can easily verify that it is provable by figuring out the following reasoning path: we can select the two premises about "John" and deduce "John is a greedy King." by combining them; we then pick the premise about "greedy King"

and conclude "John is evil." by combining it with the previous deduction. In this paper, we build an automatic system that is able to perform this kind of human-like logical reasoning.

## 3 Our LEAP Framework

We propose LEAP, an LM-based logical reasoning system that performs explicit planning. Pretrained LMs are excellent at understanding natural languages as well as fluently generating them.[1] Our LEAP system harnesses such abilities to simulate step-by-step reasoning processes that resembles how humans do logical reasoning. In this section, we will incrementally build up our full system, starting from a base system (section 3.1) to how explicit planning is integrated (sections 3.2–3.3).

### 3.1 Base System

Our base system consists of a selection model $p_{\text{sel}}$, a deduction model $p_{\text{ded}}$, and a verification model $p_{\text{ver}}$. They work together in an iterative fashion to perform multi-step reasoning like shown in Figure 1. At each step, the selection model $p_{\text{sel}}$ selects a couple of premises from the current theory. For example, at step-1 in Figure 1, it selects "eagles eat rabbits" and "rabbits are animals" from the original theory of four premises. Then the deduction model $p_{\text{ded}}$ reads the selected premises and outputs a new statement that is logically plausible given the selection. For example, at step-1 in Figure 1, it deduces "eagles eat animals". The new statement is then added to the theory (whose size increases by one) and it may be selected by $p_{\text{sel}}$ at a later step. The procedure stops if the max number of reasoning steps has been reached; otherwise, it starts a new iteration of selection and deduction. This procedure gives a reasoning path as shown in Figure 1.

We define the *proof* score of the reasoning path to be

$$f(\mathcal{T}, \mathbf{x}_0) \stackrel{\text{def}}{=} \max_{n=1,\dots,N} p_{\text{ver}}(\mathbf{x}_0 \mid \mathbf{x}_n) \in (0,1) \quad (1)$$

where theory $\mathcal{T}$ has been extended to include all the new deductions obtained through the reasoning process. Each $p_{\text{ver}}(\mathbf{x}_0 \mid \mathbf{x}_n)$ is given by the verification model and measures how likely the statement $\mathbf{x}_n$ will prove the goal: e.g., "eagles only eat animals" ($\mathbf{x}_6$) should have a lower score than "eagles are carnivores" ($\mathbf{x}_7$) since the latter means the same as the goal. The proof score $f(\mathcal{T}, \mathbf{x}_0)$ can be regarded as the system's belief that the theory proves the goal.

How do we define the verification score $p_{\text{ver}}(\mathbf{x}_0 \mid \mathbf{x}_n)$? We utilize a pretrained DeBERTa model (He et al., 2021) that was fine-tuned on the standard MNLI language inference dataset (Williams et al., 2018). For a statement $\mathbf{x}_n$ and goal $\mathbf{x}_0$, we define the verification score $p_{\text{ver}}(\mathbf{x}_0 \mid \mathbf{x}_n)$ to be the DeBERTa probability that $\mathbf{x}_n$ *entails* $\mathbf{x}_0$. It is a reasonable estimate for the probability that $\mathbf{x}_n$ *proves* $\mathbf{x}_0$.

---

[1] We use "language model" broadly to refer to multiple types of language representation models including encoder-only, decoder-only, and encoder-decoder models.

Our system is general: the selection and deduction models can be any pretrained decoder-only or encoder-decoder models, including the small models whose parameters we could update and the huge models that we could only use as blackboxes. In section 4, we will discuss some specific model choices as well as how to transfer them to our logical reasoning problem. Generally, we only require that

- the selection model $p_{\text{sel}}$ can propose multiple multi-premise selections given the theory $\mathcal{T}$ and assign a score to each of them. For a multi-premise selection $\mathbf{s}$ (e.g., $\mathbf{s} = \mathbf{x}_2\mathbf{x}_3$), we denote the score to be $p_{\text{sel}}(\mathbf{s} \mid \mathcal{T}, \mathbf{x}_0)$, or $p_{\text{sel}}(\mathbf{s})$ for short.

- the deduction model $p_{\text{ded}}$ can draw multiple deductions given a selection $\mathbf{s}$ and assign a score to each of them. For a deduction $\mathbf{x}$, we denote its score to be $p_{\text{ded}}(\mathbf{x} \mid \mathbf{s})$.

So far, we have been assuming that we select the highest scored selection and deduction at each step (e.g., in Figure 1 and at the beginning of this section). But this kind of one-best decoding tends to be short-sighted: there may be multiple possible reasoning paths to proving the goal; some may be better than the others (e.g., they are shorter) but they may not appear to be promising at the early steps; such reasoning paths may be missed by one-best decoding. Therefore, we develop an improved decoding method that resembles beam search (Jurafsky and Martin, 2000).

**Beam-search-style inference.** We maintain a buffer $\mathcal{B}$ of maximum size $B$ which can host at most $B$ ongoing reasoning paths, which we think are the most promising and will eventually prove the goal. Each of ongoing path tracks its proof score $f$ as well as its log-probability $g$ under our system. Both $f$ and $g$ get updated as the path progresses, which we will explain shortly. It also tracks its initial theory as well as its selections and deductions; the initial theory and the deductions form the extended (or current) theory. As long as we haven't reached the maximum number of steps, we keep expanding each ongoing path in the buffer. Each step of expansion includes a selection step followed by a deduction step. At the selection step, we do the following:

- For each ongoing path, we find its top $B$ most probable selections $(u_1, \mathbf{s}_1), \ldots, (u_B, \mathbf{s}_B)$ where $u_b$ is the log-probability $\log p_{\text{sel}}(\mathbf{s}_b)$. Each selection expands its ongoing path and updates its $g$ score by $g \leftarrow g + u_b$.

- Now we have $B^2$ extended paths and let the buffer $\mathcal{B}$ only keep $B$ of them which are most probable under the system (i.e., those with the highest $g$).

At the deduction step, we follow a similar procedure:

- For each ongoing path, we draw its top $B$ most probable deductions $(v_1, \mathbf{y}_1), \ldots, (v_B, \mathbf{y}_B)$ conditioned on the most recent selection $\mathbf{s}$; $v_b$ is the log-probability

$p_{\text{ded}}(\mathbf{y}_b \mid \mathbf{s})$ under deduction model $p_{\text{ded}}$. Each deduction expands the ongoing path: it updates the scores by $g \leftarrow g + v_b$ and $f \leftarrow \max\{f, p_{\text{ver}}(\mathbf{x}_0 \mid \mathbf{y}_b)\}$.

- Now we end up with $B^2$ extended paths and only keep $B$ of them which have the highest $g$.

In the end, we return the reasoning path with the highest proof score $f$: intuitively, among all the choices that are probable under the selection and deduction models, we'd like to pick what's most likely to actually prove the goal. This method becomes one-best decoding if we set $B = 1$.

Appendix B.1 has more details of the base system, including pseudocode for inference (Algorithms 1–3).

**Relations to formal logic systems.** Our base system resembles a rule-based system and the inference method is like a combination of the forward and backward chaining algorithms (Russell and Norvig, 2010). Each deduction step extends the theory by deducing new facts from the existing facts and rules, which resembles the forward chaining algorithm. Each selection step is conditioned on the goal, which resembles the backward chaining algorithm. However, the forward and backward algorithms can not handle the theories that have non-definite clauses like "Either John or Richard is evil."; our method doesn't have that limitation.

### 3.2 Improvement-A: Inference with Planning

The inference method in section 3.1 lacks *planning*. While expanding each ongoing path, the selections and deductions are ranked by their scores $u$ and $v$ that are only conditioned on the previous selections and deductions. However, the selections and deductions that appear to be promising may not actually lead to the *future* steps that are able to prove the goal. In this section, we propose an improved inference method that ranks the selections and deductions by explicit planning. We refer to the improved version as System-A.

**Planning for selection.** At each selection step, we expand each ongoing reasoning path with $B$ selections given by the no-planning method, and let the buffer $\mathcal{B}$ keep $B$ of the $B^2$ extended paths with the highest scores. The key improvement is: we redefine the score such that it reflects not only the probability of the selection under the model $p_{\text{sel}}$ but also the quality of the future steps that the selection leads to.

Precisely, we redefine $u = \log p_{\text{sel}}(\mathbf{s}) + \alpha \Delta u$ where $\alpha$ is a tunable hyperparameter and $\Delta u$ is a future-specific correction term that we can compute after rolling out some imaginary future deductions. For a possible selection $\mathbf{s}$, we call the base one-best decoding method (section 3.1) to roll out $D$ steps of future deductions $\tilde{\mathbf{y}}_1, \ldots, \tilde{\mathbf{y}}_D$. Then we obtain $p_{\text{ver}}(\mathbf{x}_0 \mid \tilde{\mathbf{y}}_d)$—which evaluates how likely each rolled-out deduction may entail the goal—and compute the correction term by $\Delta u \overset{\text{def}}{=} \max_d \log p_{\text{ver}}(\mathbf{x}_0 \mid \tilde{\mathbf{y}}_d)$. Note that $\Delta u$ is the logarithm of the proof score defined on the rolled-out future

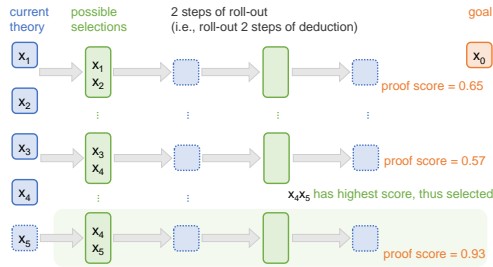 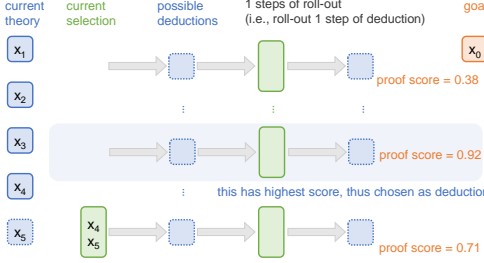

(a) Planning for selection.   (b) Planning for deduction.

Figure 2: An illustration of explicit planning at the 2nd selection and deduction step of the full procedure in Figure 1.

reasoning path. Intuitively, a higher $\Delta u$ means that this future reasoning path is more likely to prove the goal. In the end, we obtain $B$ selections with updated scores $(u_1, \mathbf{s}_1), \ldots, (u_B, \mathbf{s}_B)$ for each ongoing path.

This improved subroutine is illustrated in Figure 2a. Its pseudocode is Algorithm 10 in Appendix B.5.

**Planning for deduction.** At each deduction step, we expand each ongoing reasoning path with $B$ deductions given by the no-planning method, and let the buffer $\mathcal{B}$ keep $B$ of the extended paths with the highest scores. Similar to the planning-based selection step, the key improvement is the refined definition of the score, which reflects not only the probability of the deduction under the model $p_{\mathrm{ded}}$ but also the quality of its future steps.

Precisely, we first draw $B$ most probable deductions $(v_1, \mathbf{y}_1), \ldots, (v_B, \mathbf{y}_B)$ under the model $p_{\mathrm{ded}}$. Then we edit the score $v_b \leftarrow v_b + \beta \Delta v_b$ where $\beta$ is a tunable hyperparameter and $\Delta v$ is a future-specific correction similar to $\Delta u$. For each possible deduction $\mathbf{y}_b$, we call the no-planning one-best decoding method to roll out $D$ steps of future deductions $\tilde{\mathbf{y}}_{b,1}, \ldots, \tilde{\mathbf{y}}_{b,D}$. Then we compute $\Delta v_b \stackrel{\text{def}}{=} \max_d \log p_{\mathrm{ver}}(\mathbf{x}_0 \mid \tilde{\mathbf{y}}_{b,d})$. In the end, we obtain $B$ deductions with updated scores $(v_1, \mathbf{y}_1), \ldots, (v_B, \mathbf{y}_B)$ for each ongoing path.

This improved subroutine is illustrated in Figure 2b. Its pseudocode is Algorithm 11 in Appendix B.5.

**The full method.** Except for the score definitions, the planning-based inference method looks the same as the no-planning method: the top selections and deductions will expand their ongoing paths and update their scores $f$ and $g$; the buffer will only keep $B$ paths with the highest $g$. But the planning-based method will tend to end up with a different set of reasoning paths than the no-planning method since the scores have been affected by the roll-outs. The full inference algorithm is Algorithm 1 in Appendix B.1: when $D \geq 1$, it does explicit planning; when $D = 0$, it doesn't roll out future steps and becomes the no-planning method.

**System 1 vs. System 2 reasoning.** According to the "dual process" theories of reasoning (Evans, 2003), human cognition can be thought of as an interplay between a fast and intuitive "System 1" and a slow but analytical "System 2". Given enough time, System 2 can analyze the default behavior of System 1 and override it if neces-

sary. In analogy to this process, our base system can be considered as System 1, while the advanced planning-based system is like System 2, which requires more computation but performs more deliberative reasoning.

Precisely, at each step of reasoning, the no-planning base system needs $3B$ operations (i.e., select, deduce, and verify). In contrast, the planning-based inference needs $3B + 3B^2 D + 3B^2 D$ operations: for each ongoing reasoning path in the buffer, we need to examine its $B$ possible expansions (selection or deduction), and roll out $D$ future steps (via one-best decoding) for each expansion. Overall, the planning-based system consumes $1 + 2BD$ times of computation. Fortunately, our implementation is efficient because of careful tensorization and parallelism; please see section 6.1 for an analysis of its actual walk-clock time.

### 3.3 Improvement-B: Refined Verification Model

The key limitation of the planning method is that it may exploit the pretrained verification model $p_{\mathrm{ver}}$ such that the final proof score $f(\text{theory}, \text{goal})$ is inflated: this method keeps ongoing paths that have high $p_{\mathrm{ver}}(\text{goal} \mid$ possible future deductions). This will result in a high rate of false positive: even when the goal is not provable, explicit planning will still try its best to find out the reasoning paths that have high proof scores; a high proof score will then fool the system itself to believe that this goal is provable. This issue is illustrated in our experiments (see Figure 5c and related analysis in section 6.1). In this section, we propose to resolve this issue by refining our verification model. We refer to this version of our LEAP system as System-B.

Our method is to tune the verification model $p_{\mathrm{ver}}$ such that $p_{\mathrm{ver}}(\text{goal} \mid \text{deduction})$ is low when the deduction can *not* prove the goal. Technically, given a theory $\mathcal{T}$ and a *non-provable* goal $\bar{\mathbf{x}}_0$, we first call our planning-based method to find a reasoning path that tries to prove $\bar{\mathbf{x}}_0$, and then make $p_{\mathrm{ver}}(\bar{\mathbf{x}}_0 \mid \bar{\mathbf{y}})$ to be low for each deduction $\bar{\mathbf{y}}$ in the reasoning path. Precisely, we locally minimize $\ell$:

$$\log p_{\mathrm{ver}}(\bar{\mathbf{x}}_0 \mid \bar{\mathbf{y}}) - \log\left(p_{\mathrm{ver}}(\bar{\mathbf{x}}_0 \mid \bar{\mathbf{y}}) + p_{\mathrm{ver}}(\mathbf{x}_0 \mid \mathbf{y})\right) \tag{2}$$

where $\mathbf{x}_0$ is a provable goal and $\mathbf{y}$ is a deduction in a reasoning path that actually proves $\mathbf{x}_0$. This objective $\ell$ is a typical contrastive learning objective (Ma and Collins, 2018). In our setting, it means: if we are given

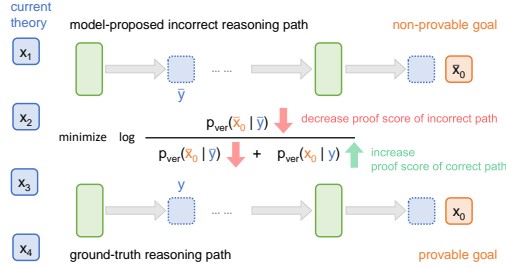

Figure 3: Illustration of our contrastive learning framework for refining verification model.

a non-provable goal $\bar{\mathbf{x}}_0$ paired with a model-proposed reasoning path as well as a provable goal $\mathbf{x}_0$ paired with a correct reasoning path, our verification model $p_{\text{ver}}$ should learn to correctly judge that "$\bar{\mathbf{x}}_0$ proved by path of $\bar{\mathbf{y}}$" is *less likely* than "$\mathbf{x}_0$ proved by path of $\mathbf{y}$". This framework is illustrated in Figure 3.

Additionally, we augment the loss $\ell$ with

$$\Omega = - p_{\text{ver}}^-(\mathbf{x}_0 \mid \mathbf{y}) \log p_{\text{ver}}(\mathbf{x}_0 \mid \mathbf{y}) \tag{3a}$$
$$- \left(1 - p_{\text{ver}}^-(\mathbf{x}_0 \mid \mathbf{y})\right) \log \left(1 - p_{\text{ver}}(\mathbf{x}_0 \mid \mathbf{y})\right) \tag{3b}$$

where $p_{\text{ver}}^-$ is the pretrained verification model used in sections 3.1 and 3.2. It is the KL-divergence (minus $H(p_{\text{ver}}^-)$, which is a constant wrt. model parameters) between the pretrained and tuned verification models, and minimizing it aims to prevent the tuned model from deviating too much from the pretrained. This is desirable since the pretrained model already enjoys a high rate of true positive for provable goals; see results in Figure 5b and relevant analysis in section 6.1.

Technical details (including visualization) about the verification model are in Appendix B.2.

# 4 Small and Large Model Versions

Now we introduce two specific versions of our proposed framework: the small language model (SLM) version that uses pretrained T5 (Raffel et al., 2020) and the large language model (LLM) version that utilizes GPT-3.5.

## 4.1 SLM Version

Our SLM version adapts pretrained T5 models (Raffel et al., 2020) to be the selection and deduction models. We use the T5-small instance (from Huggingface) that has only 60M parameters because we would like to investigate how well a very small system will work in practice. Shortly in section 6, we will see that this small system works very well.

Given a theory $\mathcal{T}$ and a goal $\mathbf{x}_0$, the selection T5 model reads them as input and produces the probability $p_{\text{sel}}(\mathbf{x}_n \mid \mathcal{T}, \mathbf{x}_0)$ that each premise $\mathbf{x}_n$ is selected in the attempt to prove the goal $\mathbf{x}_0$. Then we can use these probabilities to compute the probability $p_{\text{sel}}(\mathbf{s} \mid \mathcal{T}, \mathbf{x}_0)$ that a multi-premise combination $\mathbf{s}$ (e.g., $\mathbf{s} = \mathbf{x}_2\mathbf{x}_4$) is

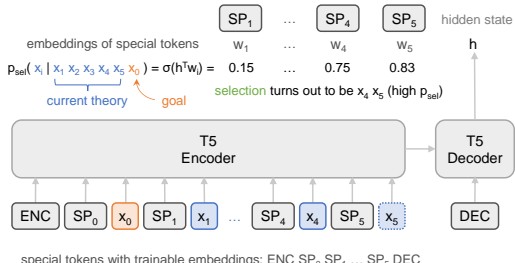

(a) A selection step. The T5 encoder reads special tokens, the goal $\mathbf{x}_0$, and the theory $\mathcal{T}$. The decoder computes $p_{\text{sel}}(\mathbf{x}_n \mid \mathcal{T}, \mathbf{x}_0) \stackrel{\text{def}}{=} \sigma(\mathbf{h}^\top \mathbf{w}_n)$ where $\mathbf{w}_n$ is the embedding of special token $SP_n$.

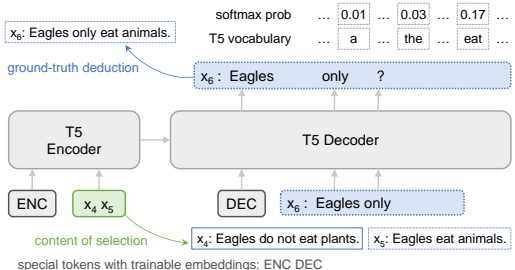

(b) A deduction step. The T5 encoder reads special tokens and the selection $\mathbf{s} = \mathbf{x}_4 \; \mathbf{x}_5$ and generates a deduction autoregressively. It is currently trying to find the token after "only", and "eat" wins.

Figure 4: An illustration of how the SLM selection and deduction models in the example procedure of Figure 1.

selected:[2]

$$\prod_{n : \mathbf{x}_n \in \mathbf{s}} p_{\text{sel}}(\mathbf{x}_n \mid \mathcal{T}, \mathbf{x}_0) \prod_{n : \mathbf{x}_n \notin \mathbf{s}} (1 - p_{\text{sel}}(\mathbf{x}_n \mid \mathcal{T}, \mathbf{x}_0))$$

Then finding the most probable selection is to choose the premises $\mathbf{x}_n$ that have $p_{\text{sel}}(\mathbf{x}_n \mid \mathcal{T}, \mathbf{x}_0) > 0.5$.[3] This procedure is illustrated in Figure 4a.

Give a selection $\mathbf{s}$, the deduction T5 model reads $\mathbf{s}$ and produces a logical deduction $\mathbf{y}$ one token after another. The probability of $\mathbf{y}$ under the model is $p_{\text{ded}}(\mathbf{y} \mid \mathbf{s})$. Figure 4b shows a deduction step.

Training the SLM version requires a corpus of theories and goals as well as their ground-truth reasoning paths. The selection steps are training examples for $p_{\text{sel}}$; the deduction steps are training examples for $p_{\text{ded}}$. Taking Figure 1 as an example, the selection training data (green background) is

- $\mathcal{T} = \{\mathbf{x}_1, \mathbf{x}_2, \mathbf{x}_3, \mathbf{x}_4\}$ and $\mathbf{s} = \mathbf{x}_2 \, \mathbf{x}_3$
- $\mathcal{T} = \{\mathbf{x}_1, \mathbf{x}_2, \mathbf{x}_3, \mathbf{x}_4, \mathbf{x}_5\}$ and $\mathbf{s} = \mathbf{x}_4 \, \mathbf{x}_5$
- $\mathcal{T} = \{\mathbf{x}_1, \mathbf{x}_2, \mathbf{x}_3, \mathbf{x}_4, \mathbf{x}_5, \mathbf{x}_6\}$ and $\mathbf{s} = \mathbf{x}_1 \, \mathbf{x}_6$

and the deduction training data (blue background) is

- $\mathbf{s} = \mathbf{x}_2 \, \mathbf{x}_3$ and new statement $\mathbf{y} = \mathbf{x}_5$

---

[2] We treat each $\mathbf{x}_n$ independently.

[3] For each $\mathbf{x}_n$, if $p_{\text{sel}} > 0.5$, we will have $p_{\text{sel}} > 1 - p_{\text{sel}}$. That is, including it in $\mathbf{s}$ will increase the probability of $\mathbf{s}$.

- $\mathbf{s} = \mathbf{x}_4\ \mathbf{x}_5$ and new statement $\mathbf{y} = \mathbf{x}_6$

- $\mathbf{s} = \mathbf{x}_1\ \mathbf{x}_6$ and new statement $\mathbf{y} = \mathbf{x}_7$

The training objectives for the selection model $p_{\text{sel}}$ and deduction model $p_{\text{ded}}$ are $\log p_{\text{sel}}(\mathbf{s} \mid \mathcal{T}, \mathbf{x}_0)$ and $\log p_{\text{ded}}(\mathbf{y} \mid \mathbf{s})$, respectively.

Appendix B.3 includes more details about the SLM version (e.g., pseudocode for training and inference).

### 4.2 LLM Version

Our LLM uses GPT-3.5-turbo as the selection and deduction models. GPT-3.5 is the current largest and state-of-the-art language model that we have access to. We instruct GPT-3.5 to perform selection and deduction by few-shot prompting; please see Appendices B.4 and C.6 for technical details and the prompts used in our experiments. This is similar to the selection-inference framework proposed by Creswell et al. (2023) except that we request GPT-3.5 to propose *multiple* possible selections and deductions at each step. This design allows us to perform explicit planning for each possible selection and deduction and then choose the best option based on planning. Since GPT-3.5 doesn't give the values of the probabilities $p_{\text{sel}}$ and $p_{\text{ded}}$, we set $u = v = 0$ in the inference methods, conditioning the selection and deduction entirely on the planning signals. The proof score $f$ is still given by the DeBERTa verification model that we introduced in section 3.

## 5 Related Work

Reasoning has been a long-standing research topic in natural language processing. For a long time, the majority of research in this direction has been focused on simple tasks such as single-sentence language inference (Bernardi, 2002; Zamansky et al., 2006; MacCartney and Manning, 2009; Angeli et al., 2016; Hu et al., 2020; Chen et al., 2021) and single-step commonsense inference (Rajani et al., 2019; Latcinnik and Berant, 2020; Shwartz et al., 2020).

Recently, there has been an increasing research interest in the more complex problem of multi-step logical reasoning, which we study in this paper. Saha et al. (2020), to the best of our knowledge, is the first to propose an interpretable LM-based model for this problem. They and Tafjord et al. (2021) work on synthesized data of limited language variability. The LM-based system proposed by Bostrom et al. (2022) has an architecture similar to the SLM version of our base system except that their inference is one-best decoding without planning and their deduction model is trained with extra data collected by Bostrom et al. (2021). The selection-inference system of Creswell et al. (2023) is similar to the LLM version of our base system but their selection and deduction models are few-shot-prompted GPT-3; we compare with them in section 6.3. Liu et al. (2022) also use a similar architecture which they train by reinforcement learning. Weir and Van Durme (2022) embed LMs into a backward chaining framework, achieving strong performance in scientific reasoning. Our main contribution is complementary to the previous work: we integrate explicit planning into LM-based reasoning systems and design a training method to mitigate the model exploitation issue that arises in planning. Our system is a kind of general model programs (Dohan et al., 2022)—especially those with verification models (Cobbe et al., 2021)—which use language models inside as probabilistic programs and apply disparate inference algorithms to the models. Other kinds of approaches to use LMs for reasoning include training discriminative models (Clark et al., 2020; Picco et al., 2021; Ghosal et al., 2022; Zhang et al., 2023), prompting GPT-3 with spelled-out reasoning procedure (Wei et al., 2022; Talmor et al., 2020), and distilling GPT-3.5 (Fu et al., 2023).

Another straightforward approach for text-based logical reasoning is to first translate natural language statements into formal logic expressions and then use a formal logic inference engine (Weber et al., 2019; Levkovskyi and Li, 2021; Nye et al., 2021; Lu et al., 2022b; Betz and Richardson, 2022). We tried this approach in our experiments; please see Appendix C.3 for details.

Another research area related to multi-step logical reasoning is to reason over graph-structured data. A popular kind of graph is knowledge graphs, i.e., relational graphs over symbolic tuples (Lao and Cohen, 2010; Wang et al., 2013; Neelakantan et al., 2015; Cohen et al., 2017; Xiong et al., 2017; Chen et al., 2018; Das et al., 2018). Another kind of graph is built by linking texts via lexical overlap or hyperlink connections (Welbl et al., 2018; Yang et al., 2018; Khot et al., 2020, 2021). Methods in this area involve multi-step navigation through graphs. But they rely on pre-defined symbolic and relational structures, thus not directly applicable to our setting. Additionally, recent research (Chen and Durrett, 2019; Min et al., 2019) shows that optimizing the performance on these datasets is not well aligned to improving the models' fundamental reasoning abilities.

## 6 Experiments

We carried out a diverse set of experiments that can demonstrate the effectiveness of our proposed methods. We implemented our methods with PyTorch (Paszke et al., 2019) and Transformers (Wolf et al., 2020). Our code is at https://github.com/cindermond/leap.

### 6.1 SLM Experiments on Entailment Bank

We first trained and evaluated our SLM version on the standard benchmark Entailment Bank (Dalvi et al., 2021) dataset. This dataset is a corpus of human-annotated (theory, provable goal, reasoning path) tuples, including the example in Figure 1. It uses informal language, which closely aligns with how humans engage in logical reasoning during everyday conversations. This dataset has two versions: in Version-I, for each pair of theory and goal, all the premises have to be used to prove the goal; in Version-II, each theory includes a few distractors that are not useful for proving the goal.

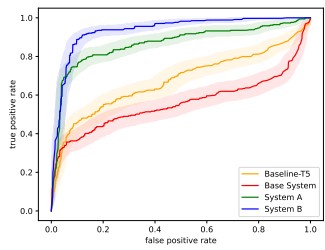
(a) ROC curves.

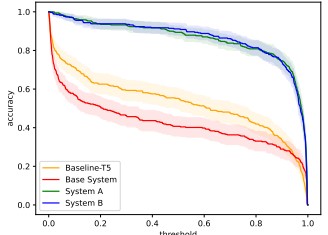
(b) Acc curves on positive examples.

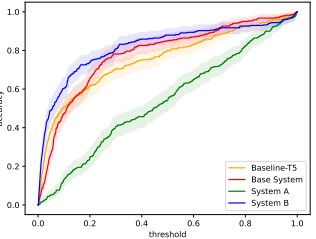
(c) Acc curves on negative examples.

Figure 5: Test results with 95% bootstrap confidence intervals (CFs) on Entailment Bank Version-I.

| METHOD | AUROC | AUACC$_{POS}$ | AUACC$_{NEG}$ | F1 |
|---|---|---|---|---|
| BASELINE-T5 | 0.67 (0.63, 0.71) | 0.53 (0.49, 0.57) | 0.75 (0.72, 0.78) | 0.62 (0.59, 0.64) |
| BASE SYSTEM | 0.56 (0.51, 0.60) | 0.42 (0.38, 0.47) | 0.78 (0.76, 0.81) | 0.67 (0.67, 0.67) |
| SYSTEM A | 0.87 (0.84, 0.89) | 0.86 (0.84, 0.89) | 0.54 (0.50, 0.57) | 0.82 (0.80, 0.84) |
| SYSTEM B | **0.94** (0.92, 0.95) | **0.87** (0.84, 0.89) | **0.82** (0.79, 0.85) | **0.89** (0.87, 0.91) |
| RULETAKER | **0.90** (0.88, 0.93) | **0.91** (0.88, 0.94) | **0.73** (0.69, 0.77) | 0.84 (0.83, 0.86) |
| NEURALUNIF | 0.72 (0.68, 0.76) | 0.56 (0.56, 0.57) | 0.49 (0.48, 0.50) | 0.72 (0.71, 0.74) |
| GPT-3 (0-SHOT) | - | - | - | **0.89** |

Table 1: Test results with 95% bootstrap CFs on Entailment Bank Version-I.

We trained the models on Version-I training data, but evaluated them on both Version-I and Version-II test data. Experiment details are in Appendix C, including data statistics (Table 5) and training details (e.g., hyperparameter tuning in Appendix C.2).

**Evaluation-I: binary classification.** We evaluated the abilities of the systems to classify provable and non-provable goals. For this purpose, we gave a non-provable goal to each dev and test theory by selecting it from other (theory, goal, reasoning path) samples. The selection is adversarial: we tuned a pretrained T5 model to generate a provable goal given a theory; for each theory $\mathcal{T}$, we looped over all the goals in the dataset that are guaranteed to be not provable under $\mathcal{T}$, and chose the one that the T5 thinks is the most probable given $\mathcal{T}$ (see details in Appendix C).

For each given theory $\mathcal{T}$ and goal $\mathbf{x}_0$, we let the system generate a reasoning path that tries to prove the goal, and obtain the proof score $f(\mathcal{T}, \mathbf{x}_0)$ of the path. Given a threshold $\tau \in (0, 1)$, we say "$\mathbf{x}_0$ is provable" if $f(\mathcal{T}, \mathbf{x}_0) \geq \tau$ and "$\mathbf{x}_0$ is not provable" otherwise. For a systematic investigation, we varied $\tau$ and plot a receiver operating characteristic (ROC) curve for each system; the larger the area under ROC curve (AUROC) is, the better the system is.

The ROC curves are shown in Figure 5a: our LEAP System-A and System-B substantially and significantly outperform the base system and a T5 model (trained on generating goals given theories); System-B further significantly outperforms System-A. Surprisingly, our base system underperforms the T5 model even though it has learned to spell out its reasoning steps which we expect to help the classification.

Figure 5b and Figure 5c show the results broken down into the accuracies on the provable goals and non-provable goals, respectively. On provable goals, the accuracy is the number of true positive divided by the total number of test cases; on non-provable goals, the accuracy is the number of true negative divided by the total number of test cases. As we can see, System-A works very well on the provable goals, but performs poorly on the non-provable goals. That is because System-A exploits the verification model by explicit planning: as we have discussed in section 3.3, the proof scores given by System-A tend to be high, thus yielding a high rate of false positive. System-B works well on both provable and non-provable goals: the refined verification model $p_{\mathrm{ver}}$ successfully avoided being exploited by planning. Actual values of the areas under curves are shown in Table 1: AUACC$_{\mathrm{pos}}$ and AUACC$_{\mathrm{neg}}$ correspond to the curves in Figure 5b and Figure 5c, respectively. The F1 numbers were computed as follows: we chose an optimal threshold $\tau$ by maximizing the F1 score on the development set, and then computed F1 on the test set according to the chosen $\tau$.

For a comprehensive evaluation, we also compared with three other kinds of methods: GPT-3-davinci with 0-shot prompting, RuleTaker (Clark et al., 2020), and Neural Unification (Picco et al., 2021). GPT-3 achieves a strong F1 of 0.89, and our System-B performs as well as this strong model. RuleTaker is a discriminative method, training a RoBERTa (Liu et al., 2019) to perform logical reasoning as binary classification (provable or not). Neural Unification is also a discriminative method but has a different architecture than RuleTaker. It requires more sophisticated annotation and preparation of the training data than RuleTaker and our methods. Neither of them spells out a reasoning process. For these methods, we matched their numbers of trainable parameters with our methods for a fair comparison. Overall, RuleTaker performs better than our System-A but worse than System-B. Neural Unification performs worse than

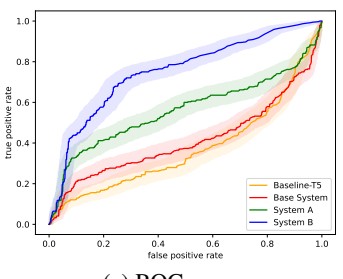 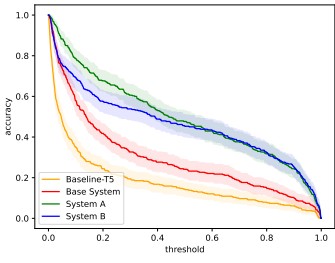 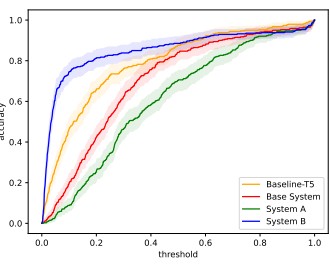

| (a) ROC curves. | (b) Acc curves on positive examples. | (c) Acc curves on negative examples. |

Figure 6: Test results with 95% bootstrap CFs on Entailment Bank Version-II.

RuleTaker and our System-A. Note that these results are orthogonal to our main finding that explicit planning is helpful for text-based multi-step logical reasoning.

**Analysis-I: robustness to size of training data.** We also trained the models with (randomly sampled) 50% of the training data, and evaluated them on the same test set. It turns out that our System-B still performs the best; see Figure 8 (which looks boringly similar to Figure 5) in Appendix C.4 for details.

**Analysis-II: About the regularization in equation (3).** We compared the system B with and without the regularization term $\Omega$: without $\Omega$, System-B only achieves AUROC $= 0.79$ (AUROC$_{\text{pos}} = 0.68$ and AUROC$_{\text{neg}} = 0.65$), worse than System-A. We also evaluated the tuned verification models on the MNLI dataset (on which they were fine-tuned) and found that: the model tuned without $\Omega$ only achieved 62.0% accuracy; the model tuned with $\Omega$ achieved 91.4% accuracy, almost as good as it originally was (91.7%). It means that the regularization term indeed helps the verification model preserve its ability to judge the entailment relationship.

**Analysis-III: Robustness to distractors.** We investigated the robustness of the systems to distractors by evaluating them on Version-II test data. Note that they were only trained on Version-I training data. As shown in Figure 6, all the systems perform worse than they did on Version-I test data, but the performance drop of our systems is much smaller than that of the T5 model. It means that our systems are more robust to the distractors. That is perhaps because our systems explicitly spell out their reasoning steps and explicit planning can help the systems (A and B) focus on the premises that are actually relevant to the goal at each selection step.

**Analysis-IV: About model size and denoising.** To examine the effect of model size, we reran the main experiments with T5-small (60M) replaced by T5-base (220M): using a larger model achieved a consistently stronger performance; our planning-based systems still significantly outperform the base system. We also experimented with denoising training of the selection and deduction models: every time we used a training example, we randomly permuted the input statements. The denoising training led to a better generalization to the

| METHOD | VERSION-I | VERSION-II |
|---|---|---|
| BASELINE-T5 | 0.60 (0.55, 0.65) | 0.20 (0.16, 0.24) |
| BASE SYSTEM | 0.46 (0.41, 0.52) | 0.29 (0.25, 0.34) |
| SYSTEM A | 0.80 (0.76, 0.84) | 0.44 (0.39, 0.49) |
| SYSTEM B | **0.88** (0.85, 0.92) | **0.63** (0.58, 0.68) |
| RULETAKER | 0.83 (0.79, 0.87) | 0.73 (0.68, 0.77) |
| NEURALUNIF | 0.62 (0.55, 0.69) | 0.62 (0.57, 0.67) |
| GPT-3 (0-SHOT) | 0.72 | 0.20 |
| GPT-3 (5-SHOT) | 0.97 | 0.96 |
| GPT-3 (COT) | **0.98** | **0.98** |

Table 2: Test accuracy with 95% bootstrap CFs in multiple-choice QA. Accuracy of random guess is 25%.

evaluation settings with distractors. We also found that training with distractors (i.e., using Verstion-II training data) significantly improved the results. Detailed results and analysis are in Table 6 and Table 7 of Appendix C.4.

**Analysis-V: About buffer size.** The buffer size $B$ is a tunable hyperparameter. In our experiments, we chose $B = 5$, a common choice in text generation. A pilot experiment with $B \in \{2, 3, 5, 10\}$ showed that: a smaller $B$ tends to slightly decrease the accuracy on positive samples, but increase it on negative samples; a larger $B$ tends to slightly increase the accuracy on positive samples, but decreases it on negative samples; overall, there are only tiny changes in AUROC, which depends on accuracies on both kinds of samples.

**Analysis-VI: Computation Cost.** In our experiments, we used $B = 5$ and $D = 2$, i.e., a buffer size of 5 and a roll-out depth of 2. According to the theoretical analysis in section 3.2, the planing-based inference should be $1 + 2BD = 21$ times slower than the no-planning method. In practice, it takes an average of 2.8 seconds for the no-planning method to work on a theory-goal pair from Entailment Bank. For the planning-based inference, it takes an average of 31 seconds, only 11 times slower. The implementation is faster than the theoretical analysis thanks to tensorization and parallelism.

**Evaluation-II: Multiple-Choice QA.** We further evaluated the systems in a multiple-choice question answering (QA) setting. Particularly, given a theory $\mathcal{T}$ in Entailment Bank, each system is asked to select the provable goal from four choices $\{\mathbf{x}_0^{(1)}, \mathbf{x}_0^{(2)}, \mathbf{x}_0^{(3)}, \mathbf{x}_0^{(4)}\}$: one of them is the ground-truth provable goal while the

| METHOD | ACC |
|---|---|
| BASE SYSTEM | 0.68 (0.65, 0.71) |
| SYSTEM A | 0.84 (0.82, 0.87) |
| SYSTEM B | **0.85** (0.83, 0.87) |

Table 3: Dev accuracy with 95% bootstrap CFs on QASC.

| METHOD | DEPTH=1 | DEPTH=3 | DEPTH=5 |
|---|---|---|---|
| COT | 0.71 (0.62, 0.80) | **0.57** (0.47, 0.67) | 0.52 (0.42, 0.62) |
| SI | 0.88 (0.81, 0.94) | 0.51 (0.41, 0.61) | 0.45 (0.35, 0.55) |
| SYSTEM A | **0.90** (0.84, 0.95) | 0.55 (0.45, 0.65) | **0.55** (0.45, 0.65) |

Table 4: Accuracy with 95% bootstrap confidence intervals on PrOntoQA. The "depth" denotes the number of ground-truth reasoning steps.

others are negative choices selected by a tuned T5.

We took the systems trained in section 6.1 and evaluated them on the Version-I and Version-II of this multiple-choice task: in the Version-II setting, each theory has a few distractors, so it is more challenging than Version-I. For each theory, a system tries to prove each choice $\mathbf{x}_0^{(c)}$, ranks the four choices by their proof scores $f(\mathcal{T}, \mathbf{x}_0^{(c)})$, and then chooses the one with the highest score. The systems were evaluated by accuracy. As shown in Table 2, the systems behave similarly as they do on the binary classification: in both Version-I and Version-II settings, System-A and System-B perform significantly better than the baselines, and System-B significantly outperforms System-A.

We also evaluated GPT-3-davinci with 0-shot, 5-shot, and chain-of-thought (COT) prompting (Brown et al., 2020; Wei et al., 2022). The COT prompts include the ground-truth reasoning paths of the correct choices; examples are in Appendix C.5. Our full system outperforms 0-shot GPT-3, but underperforms 5-shot and COT GPT-3. Interestingly, 0-shot GPT-3 works worse than random guess when theories have distractors, which indicates the difficulty of this problem. In addition, we evaluated RuleTaker and Neural Unification, with their numbers of trainable parameters matched with our methods. In the Version-I setting, they both perform worse than our System-B and Neural Unification performs even worse than System-A. Interestingly, they seem to be more robust to distractors: in the Versition-II setting, Neural Unification performs competitive to our System-B, and RuleTaker performs significantly better than System-B. However, these methods do not generate interpretable reasoning processes.

### 6.2 SLM Experiments on QASC

We also trained and evaluated the systems on the QASC dataset (Khot et al., 2020), a multiple-choice question answering dataset where each question has eight candidate answers. Each training QA pair has two premises and a deduction, which can be used to train our deduction model. Each development QA pair has two premises so the reasoning system only needs to do a step of deduction but no selection. Test QA pairs have no premises given and one has to search through a pool of millions of statements to find the relevant premises, which is not the focus of this paper. So we only evaluated the systems on the development set. The results are in Table 3. Although this data only requires one step of reasoning, the planning-based systems still significantly outperform the base system, suggesting that explicit planning is indeed helpful for LM-based reasoning.

### 6.3 LLM Experiments on PrOntoQA

We evaluated the LLM version on the "fictional" version of the PrOntoQA dataset (Saparov and He, 2023). It is a binary classification task like Entailment Bank (section 6.1), but it is more challenging to large language models such as GPT-3.5 since its logical statements are about fictional characters (e.g., wumpus), meaning that a large model can not bypass the reasoning and draw correct conclusions by commonsense or memorization.

The main results are shown in Table 4. In all cases, our planning-based system outperforms the selection-inference (SI) method, meaning that explicit planning is consistently helpful. In most cases, our planning-based system performs better than the strong chain-of-thought (COT) prompting. We also experimented with the DeBERTa model tuned on the Entailment Bank training data (sections 3.3 and 6.1) and found that it couldn't improve the performance on PrOntoQA. Appendix C.6 includes more details about this set of experiments as well as more results.

## 7 Conclusion

In this paper, we presented LEAP, an LM-based logical reasoning system that integrates explicit planning into the inference method. We also proposed a method that learns to prevent the explicit planning from being misguided. Our proposed methods exhibit intriguing technical connections to other reasoning systems and can be likened to the deliberative System 2 in "dual process" theories of reasoning. In our experiments, our planning-based system outperforms strong baseline methods including the selection-inference method and chain-of-thought prompting. We will discuss several exciting avenues for further improvements in Appendix A.

## Acknowledgments

This work was supported by a research gift to the last author by Adobe Research. We thank the anonymous EMNLP reviewers and meta-reviewer for their constructive feedback. We thank our colleagues at UChicago and TTIC for helpful discussion. We also thank Hao Tan at Adobe Research, Yisi Sang at Apple, Benjamin Van Durme at Johns Hopkins University, and David Dohan at OpenAI for their helpful comments.

## Limitations

The main limitation of our proposed framework is that it requires more computation than the baseline methods that do not perform explicit planning. As discussed in section 3.2, the no-planning methods are like the

intuitive and fast System 1 (Evans, 2003) while our methods are like the analytical and slow System 2: after all, more analysis consumes more computation and thus our framework is less energy-efficient. This limitation has inspired us to explore new methods such as bandit learning to switch between two types of systems and more efficient planning (see Appendix A).

## Ethics Statement

Our work complies with the ACL Ethics Policy. It aims to build more intelligent language-based logical reasoning systems which would have a broad positive impact to the society. For example, in our daily life, an intelligent logical reasoning system may help us verify facts and identify fake news; in legal domain, it may work as an automatic paralegal and assist lawyers with their document processing and decision making; in education, it may help students reason about their mistakes and improve learning experience. Meanwhile, our methods share the same risks as other machine learning methods, such as misusage, containing data bias, and suffering from adversarial attacks. However, this paper is orthogonal to the research efforts to mitigate these issues.

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

## A Future Extensions

Our experiments have inspired us to explore several exciting avenues for further improvements.

The first is to jointly refine the selection, deduction, and verification models. In this paper, we have already shown that adversarially refining the verification model will significantly improve the performance. So a natural next step is to adversarially refine the selection and deduction models in response to the updated verification model. Allowing components of a system to adversarially refine one another has been shown useful in natural language processing (Yu et al., 2019).

The second is to develop *implicit* planning methods to improve inference efficiency. In reinforcement learning, explicit planning is often only used to help learn a value function during training; during inference, calling a value function is like planning implicitly but faster than explicit planning. This kind of methods can apply to our setting. Another way to improve efficiency is to learn a bandit that could cleverly switch between the no-planning "System 1" and our planning-based "System 2" such that we only spend more computation in the more difficult cases.

Another direction is to leverage unlabeled data, i.e., data without human-annotated reasoning paths. Such data is less expensive to collect. An LM-based reasoning system may be able to benefit from (the indirect training signals of) such data by self-supervised learning.

## B Method Details

In this section, we give details of our methods.

### B.1 Reasoning Process Details

Algorithm 1 gives a detailed explanation for how our inference method works. When $D = 0$, it is the naive method. When $D \geq 1$, it is the inference with explicit planning. During selection, we constrain the model to only select two premises for a more controllable behavior. When we compute the proof score we only consider the newly generated deductions for convenience. Its effect to results is negligible since later deductions tend to more directly prove the goal.

Algorithm 2 is designed to select a set of statements from the current theory $\mathcal{T}$, with the goal of inferring $\mathbf{x}_0$. We fix the size of the selection set to 2 in our experiments, but in principle this restriction can be removed. Algorithm 3 draws $B_{\text{ded}}$ new deductions. Their SLM versions are Algorithms 5 and 6 and the LLM versions are Algorithms 8 and 9.

### B.2 Details of Tuning the Verification Model

We use the soft prompt tuning method (Lester et al., 2021): we augment the input with a few special tokens and the only trainable parameters are the embeddings of those tokens; it is illustrated in Figure 7.

Where do we get $\mathbf{x}_0$, $\mathbf{y}$, $\bar{\mathbf{x}}_0$, and $\bar{\mathbf{y}}$? Recall that we have a training corpus of theories and goals as well

---

**Algorithm 1** Reasoning (Inference) with Our System

**Hyperparam:** max number of inference steps $M$;
    depth of planning $D$ ($D = 0$ means "no planning");
    inference beam size $B_{\text{inf}}$
**Input:** theory $\mathcal{T} = \{\mathbf{x}_1, \mathbf{x}_2, \ldots, \mathbf{x}_N\}$ and goal $\mathbf{x}_0$;
    selection model $p_{\text{sel}}$, deduction model $p_{\text{ded}}$;
    verification model $p_{\text{ver}}$
**Output:** reasoning path $\mathcal{R}$ with proof score $f$

1: **procedure** INFERENCE($\mathcal{T}, \mathbf{x}_0, p_{\text{sel}}, p_{\text{ded}}, p_{\text{ver}}$)
2:    ▷ *has access to $M$, $B_{inf}$, $D$*
3:    $\mathcal{B} \leftarrow$ PriorityQueue($B_{\text{inf}}$)
4:    ▷ *max size is $B_{inf}$ ; priority is first element of tuple*
5:    $\mathcal{B}$.add($(0, \emptyset, \mathcal{T}, -\infty)$)
6:    ▷ *init with empty path and current theory*
7:    **for** $m = 1$ **to** $M$ :
8:       ▷ *do inference at each step*
9:       ▷ *selection at step $m$*
10:       $\mathcal{B}_{\text{old}} \leftarrow \mathcal{B}$; $\mathcal{B} \leftarrow$ PriorityQueue($B_{\text{inf}}$)
11:       **for** $g_b, \mathcal{R}_b, \mathcal{T}_b, f_b$ **in** $\mathcal{B}_{\text{old}}$ :
12:          ▷ *g is log-prob of path and $f$ is its proof score*
13:          $\mathcal{S}_b \leftarrow$ SELECT($\mathcal{T}_b, \mathbf{x}_0, p_{\text{sel}}$)
14:          **if** $D > 0$ :
15:             ▷ *rank selections based on $D$-step roll-outs*
16:             $\mathcal{S}_b \leftarrow$ PLANS($\mathcal{T}_b, \mathbf{x}_0, \mathcal{S}_b, p_{\text{sel}}, p_{\text{ded}}, p_{\text{ver}}$)
17:          **for** $u_k, \mathbf{s}_k$ **in** $\mathcal{S}_b$ :
18:             ▷ *add expanded path into priority queue*
19:             ▷ *priority score changes by $u_k$*
20:             $\mathcal{B}$.add($(g_b + u_k, \mathcal{R}_b + \{\mathbf{s}_k\}, \mathcal{T}_b, f_b)$)
21:             ▷ *$\mathcal{B}$ has a fixed size $B_{inf}$: if $|\mathcal{B}| > B_{inf}$*
22:             ▷ *auto-delete lowest-priority element*
23:       ▷ *deduction at step $m$*
24:       $\mathcal{B}_{\text{old}} \leftarrow \mathcal{B}$; $\mathcal{B} \leftarrow$ PriorityQueue($B_{\text{inf}}$)
25:       **for** $g_b, \mathcal{R}_b, \mathcal{T}_b, f_b$ **in** $\mathcal{B}_{\text{old}}$ :
26:          $\mathbf{s}_b \leftarrow$ the most recent selection in $\mathcal{R}_b$
27:          $\mathcal{Y}_b \leftarrow$ DEDUCE($\mathbf{s}_b, p_{\text{ded}}$)
28:          **if** $D > 0$ :
29:             ▷ *rank deductions based on $D$-step roll-outs*
30:             $\mathcal{Y}_b \leftarrow$ PLAND($\mathcal{T}_b, \mathbf{x}_0, \mathcal{Y}_b, p_{\text{sel}}, p_{\text{ded}}, p_{\text{ver}}$)
31:          **for** $v_k, \mathbf{y}_k$ **in** $\mathcal{Y}_b$ :
32:             $\mathcal{B}$.add($(g_b + v_k, \mathcal{R}_b + \{\mathbf{y}_k\}, \mathcal{T}_b + \{\mathbf{y}_k\}, f_b)$)
33:       **for** $g_b, \mathcal{R}_b, \mathcal{T}_b, f_b$ **in** $\mathcal{B}$ :
34:          $\mathbf{y}_b \leftarrow$ the most recent deduction in $\mathcal{R}_b$
35:          ▷ *if $\mathbf{y}_b$ entails $\mathbf{x}_0$ better than any prev deduction*
36:          ▷ *update proof score of path $\mathcal{R}_b$*
37:          **if** $p_{\text{ver}}(\mathbf{x}_0 \mid \mathbf{y}_b) > f_b$ : $f_b \leftarrow p_{\text{ver}}(\mathbf{x}_0 \mid \mathbf{y}_b)$
38:    ▷ *choose reasoning path with highest proof score*
39:    $b_{\max} \leftarrow \text{argmax}_b f_b$
40:    **return** $\mathcal{R}_{b_{\max}}, f_{b_{\max}}$

Figure 7: The structure of the verification model.

**Algorithm 2** Selection Subroutine

**Hyperparam:** selection beam size $B_{\text{sel}}$
**Input:** current theory $\mathcal{T} = \{\mathbf{x}_1, \ldots, \mathbf{x}_{N+m}\}$ and goal $\mathbf{x}_0$; selection model $p_{\text{sel}}$
**Output:** selections with their scores $\{(u_k, \mathbf{s}_k)\}$

1: **procedure** SELECT($\mathcal{T}, \mathbf{x}_0, p_{\text{sel}}$)
2:    ▷ *generic method for illustration only*
3:    ▷ *in practice, we call the SLM or LLM version*
4:    ▷ *see Algorithm 5 for SLM version*
5:    ▷ *see Algorithm 8 for LLM version*
6:    ▷ *has access to $B_{sel}$*
7:    ▷ *return list $\mathcal{S}$ which contains $B_{sel}$ scored selections*
8:    ▷ *each scored selection is $(u, \mathbf{s})$*
9:    ▷ *score $u$ is defined in section 3.2*
10:   **return** $\mathcal{S}$
11: **procedure** ONEBESTSELECT($\mathcal{T}, \mathbf{x}_0, p_{\text{sel}}$)
12:    ▷ *only keeps selection with highest score*
13:    $\mathcal{S} \leftarrow$ SELECT($\mathcal{T}, \mathbf{x}_0, p_{\text{sel}}$)
14:    $(u, \mathbf{s}) \leftarrow$ highest-scored element in $\mathcal{S}$
15:   **return** $\mathbf{s}$

---

**Algorithm 3** Deduction Subroutine

**Hyperparam:** deduction beam size $B_{\text{ded}}$
**Input:** current selection $\mathbf{s}$ of statements; deduction model $p_{\text{ded}}$
**Output:** deductions with their scores $\{(v_k, \mathbf{y}_k)\}$

1: **procedure** DEDUCE($\mathbf{s}, p_{\text{ded}}$)
2:    ▷ *generic method for illustration only*
3:    ▷ *in practice, we call the SLM or LLM version*
4:    ▷ *see Algorithm 6 for SLM version*
5:    ▷ *see Algorithm 9 for LLM version*
6:    ▷ *has access to $B_{ded}$*
7:    ▷ *return list $\mathcal{Y}$ which contains $B_{ded}$ scored deductions*
8:    ▷ *each scored deduction is $(v, \mathbf{y})$*
9:    ▷ *score $v$ is defined in section 3.2*
10:   **return** $\mathcal{Y}$
11: **procedure** ONEBESTDEDUCE($\mathbf{s}, p_{\text{ded}}$)
12:    ▷ *only keeps deduction with highest score*
13:    $\mathcal{Y} \leftarrow$ DEDUCE($\mathbf{s}, p_{\text{ded}}$)
14:    $(v, \mathbf{y}) \leftarrow$ element in $\mathcal{Y}$ with highest $v$
15:   **return** $\mathbf{y}$

---

as their ground-truth reasoning paths. For each pair of theory $\mathcal{T}$ and provable goal $\mathbf{x}_0$, we could randomly sample a deduction $\mathbf{y}$ from its ground-truth reasoning path. We use the goal of another training example as our non-provable goal $\bar{\mathbf{x}}_0$, call the planning-based inference method to get a reasoning path, and sample a deduction from the reasoning path as our $\bar{\mathbf{y}}$.

Algorithm 4 shows how we refine the verification model using the contrastive loss with regularization.

---

**Algorithm 4** Refining Verification Model

**Input:** provable goal $\mathbf{x}_0$ and gold reasoning path $\mathcal{R}$; non-provable goal $\bar{\mathbf{x}}_0$ and model-generated path $\bar{\mathcal{R}}$; verification model $p_{\text{ver}}$
**Output:** updated verification model $p_{\text{ver}}$

1: **procedure** REFINE($\mathbf{x}_0, \mathcal{R}, \bar{\mathbf{x}}_0, \bar{\mathcal{R}}, p_{\text{ver}}$)
2:    ▷ *refining procedure*
3:    $p_{\text{ver}}^- \leftarrow$ a copy of pretrained $p_{\text{ver}}$
4:    ▷ *sample deductions from reasoning paths*
5:    randomly draw $\mathbf{y}$ from deductions in $\mathcal{R}$
6:    randomly draw $\bar{\mathbf{y}}$ from deductions in $\bar{\mathcal{R}}$
7:    ▷ *refine verification model*
8:    $\ell \leftarrow$ LOSSVER($\mathbf{x}_0, \mathbf{y}, \bar{\mathbf{x}}_0, \bar{\mathbf{y}}, p_{\text{ver}}, p_{\text{ver}}^-$)
9:    compute $\nabla \ell$ wrt. trainable parameters $\boldsymbol{\theta}_{\text{ver}}$ of $p_{\text{ver}}$
10:   update $\boldsymbol{\theta}_{\text{ver}}$ with chosen optimization method
11:   **return** $p_{\text{ver}}$
12: **procedure** LOSSVER($\mathbf{x}_0, \mathbf{y}, \bar{\mathbf{x}}_0, \bar{\mathbf{y}}, p_{\text{ver}}, p_{\text{ver}}^-$)
13:    ▷ *contrastive loss*
14:    $\ell \leftarrow \log \frac{p_{\text{ver}}(\bar{\mathbf{x}}_0|\bar{\mathbf{y}})}{p_{\text{ver}}(\bar{\mathbf{x}}_0|\bar{\mathbf{y}}) + p_{\text{ver}}(\mathbf{x}_0|\mathbf{y})}$
15:    ▷ *compute regularization*
16:    $\ell \mathrel{-}= p_{\text{ver}}^-(\mathbf{x}_0 \mid \mathbf{y}) \log p_{\text{ver}}(\mathbf{x}_0 \mid \mathbf{y})$
17:    $\ell \mathrel{-}= (1 - p_{\text{ver}}^-(\mathbf{x}_0 \mid \mathbf{y})) \log(1 - p_{\text{ver}}(\mathbf{x}_0 \mid \mathbf{y}))$
18:   **return** $\ell$

---

### B.3 SLM Details

We give SLM details in this section.

**Selection model.** The selection model $p_{\text{sel}}$ uses a pretrained encoder-decoder model T5 (Raffel et al., 2020). The encoder reads a context string concatenating the goal $\mathbf{x}_0$ and the premises $\mathbf{x}_1, \ldots, \mathbf{x}_N$ of current theory $\mathcal{T}$; the decoder computes the probabilities $p_{\text{sel}}(\mathbf{x}_n \mid \mathcal{T}, \mathbf{x}_0)$ that each premise $\mathbf{x}_n$ is selected in the attempt to prove the goal $\mathbf{x}_0$. It is illustrated in Figure 4a: besides the statements, T5 also reads a few special tokens (ENC, SP$_0$, SP$_1$, ..., SP$_N$, DEC); its decoder gives a hidden state $\mathbf{h}$, which is involved in computing $p_{\text{sel}}(\mathbf{x}_n \mid \mathcal{T}, \mathbf{x}_0) \overset{\text{def}}{=} \sigma(\mathbf{h}^\top \mathbf{w}_n)$ where $\mathbf{w}_n$ is the embedding of SP$_n$. For training and inference efficiency, we keep the pretrained T5 frozen so the only trainable parameters of the selection model $p_{\text{sel}}$—denoted as $\boldsymbol{\theta}_{\text{sel}}$—are the embeddings of the special tokens. The pseudocode of using it for inference is in Algorithm 5.

**Deduction model.** Given the selection $\mathbf{s}$, the deduction model $p_{\text{ded}}$ produces a logical deduction $\mathbf{y}$ by combining the premises in $\mathbf{s}$. The new statement $\mathbf{y}$ is added to the theory $\mathcal{T}$ whose size is then increased by one;

therefore, for a theory of size $N$, we also denote $\mathbf{y}$ as $\mathbf{x}_{N+1}$. The deduction model $p_{\text{ded}}$ uses another pre-trained T5. As shown in Figure 4b, its encoder reads an input string concatenating the selected premises along with a few special tokens; its autoregressive decoder produces a deduction one token after another. Its trainable parameters $\boldsymbol{\theta}_{\text{ded}}$ are the embeddings of the special tokens. The pseudocode of deploying it is in Algorithm 6.

---

**Algorithm 5** Selection Subroutine for SLM
___

**Hyperparam:** selection beam size $B_{\text{sel}}$
**Input:** current theory $\mathcal{T} = \{\mathbf{x}_1, \ldots, \mathbf{x}_{N+m}\}$ and goal $\mathbf{x}_0$; prompted encoder-decoder language model $p_{\text{sel}}$
**Output:** selections with their scores $\{(u_k, \mathbf{s}_k)\}$
1: **procedure** SELECT($\mathcal{T}, \mathbf{x}_0, p_{\text{sel}}$)
2: $\quad \triangleright$ *has access to $B_{sel}$*
3: $\quad \triangleright$ *build context by concatenating hypothesis and theory*
4: $\quad \mathbf{c} = \text{SP}_0 + \mathbf{x}_0 + \text{SP}_1 + \mathbf{x}_1 + \ldots + \text{SP}_{N+m} + \mathbf{x}_{N+m}$
5: $\quad$ **for** $i = 1$ **to** $N + m$ :
6: $\quad\quad \triangleright$ *compute prob that each statement is selected*
7: $\quad\quad p_i \leftarrow p_{\text{sel}}(\text{SP}_i | \mathbf{c})$
8: $\quad \mathcal{S} \leftarrow \text{PriorityQueue}(B_{\text{sel}})$
9: $\quad \triangleright$ *max size is $B_{sel}$; priority is first element of tuple*
10: $\quad$ **for** $i = 1$ **to** $N + m$ :
11: $\quad\quad$ **for** $j = i + 1$ **to** $N + m$ :
12: $\quad\quad\quad \mathbf{s}_k \leftarrow \mathbf{x}_i + \mathbf{x}_j$
13: $\quad\quad\quad u_k \leftarrow \log p_i + \log p_j + \sum_{\ell \neq i, \ell \neq j} \log(1 - p_\ell)$
14: $\quad\quad\quad \mathcal{S}.\text{add}((u_k, \mathbf{s}_k))$
15: $\quad\quad\quad \triangleright$ *if $\mathcal{B}$ is larger than $B_{sel}$, element with*
16: $\quad\quad\quad \triangleright$ *lowest priority will be automatically deleted*
17: $\quad$ **return** $\mathcal{S}$

---

**Algorithm 6** Deduction Subroutine for SLM
___

**Hyperparam:** deduction beam size $B_{\text{ded}}$
**Input:** current selection $\mathbf{s}$ of statements; prompted encoder-decoder language model $p_{\text{ded}}$
**Output:** deductions with their scores $\{(v_k, \mathbf{y}_k)\}$
1: **procedure** DEDUCE($\mathbf{s}, p_{\text{ded}}$)
2: $\quad \triangleright$ *has access to $B_{ded}$*
3: $\quad \triangleright$ *has access to standard beam search implementation*
4: $\quad \mathcal{Y} \leftarrow \text{BEAMSEARCH}(p_{\text{ded}}, B_{\text{ded}}, \mathbf{s})$
5: $\quad \triangleright$ *assume:*
6: $\quad \triangleright$ *BEAMSEARCH gives a list of tuples $\{(v_k, \mathbf{y}_k)\}$*
7: $\quad \triangleright$ *text string $\mathbf{y}_k$ sorted in descending order of $v_k$*
8: $\quad$ **return** $\mathcal{Y}$

---

**Algorithm 7** Training for SLM
___

**Input:** theory $\mathcal{T} = \{\mathbf{x}_1, \ldots, \mathbf{x}_N\}$ and goal $\mathbf{x}_0$; reasoning path $\mathcal{R}$; verification model $p_{\text{ver}}$ selection model $p_{\text{sel}}$, deduction model $p_{\text{ded}}$
**Output:** updated models $p_{\text{sel}}$ and $p_{\text{ded}}$
1: **procedure** TRAIN($\mathcal{R}, \mathcal{T}, \mathbf{x}_0, p_{\text{sel}}, p_{\text{ded}}, p_{\text{ver}}$)
2: $\quad \triangleright$ *training method for selection and deduction models*
3: $\quad \triangleright$ *init extended theory that will include deduction*
4: $\quad \tilde{\mathcal{T}} \leftarrow \mathcal{T}$
5: $\quad$ **for** $m = 1$ **to** $|\mathcal{R}|/2$ :
6: $\quad\quad \triangleright$ *loop over each step of selection and deduction*
7: $\quad\quad \mathbf{s}_m \leftarrow m^{\text{th}}$ selection $\triangleright$ *i.e., $(2m-1)^{th}$ entry in $\mathcal{R}$*
8: $\quad\quad \mathbf{y}_m \leftarrow m^{\text{th}}$ deduction $\triangleright$ *i.e., $(2m)^{th}$ element in $\mathcal{R}$*
9: $\quad\quad \triangleright$ *train selection model*
10: $\quad\quad \ell \leftarrow \text{LOSSSEL}(\tilde{\mathcal{T}}, \mathbf{s}_m, p_{\text{sel}})$
11: $\quad\quad$ compute $\nabla \ell$ wrt. trainable params $\boldsymbol{\theta}_{\text{sel}}$ of $p_{\text{sel}}$
12: $\quad\quad$ update $\boldsymbol{\theta}_{\text{sel}}$ with chosen optimization method
13: $\quad\quad \triangleright$ *train deduction model*
14: $\quad\quad \ell \leftarrow \text{LOSSDED}(\mathbf{s}_m, \mathbf{y}_m, p_{\text{ded}})$
15: $\quad\quad$ compute $\nabla \ell$ wrt. trainable params $\boldsymbol{\theta}_{\text{ded}}$ of $p_{\text{ded}}$
16: $\quad\quad$ update $\boldsymbol{\theta}_{\text{ded}}$ with chosen optimization method
17: $\quad\quad \triangleright$ *extend theory with new deduction*
18: $\quad\quad \tilde{\mathcal{T}} \leftarrow \tilde{\mathcal{T}} + \{\mathbf{y}_m\}$
19: $\quad$ **return** $p_{\text{sel}}, p_{\text{ded}}$
20: **procedure** LOSSSEL($\mathcal{T}, \mathbf{s}, p_{\text{sel}}$)
21: $\quad \triangleright$ *construct context for selecting statements from theory*
22: $\quad \mathbf{c} \leftarrow \text{SP}_0 + \mathbf{x}_0 + \text{SP}_1 + \mathbf{x}_1 + \ldots + \text{SP}_{N+m} + \mathbf{x}_{N+m}$
23: $\quad \ell \leftarrow 0 \quad \triangleright$ *loss is negative log-likelihood of selection*
24: $\quad$ **for** $i = 1$ **to** $N + m$ :
25: $\quad\quad p_i \leftarrow p_{\text{sel}}(\text{SP}_i | \mathbf{c}) \quad \triangleright$ *prob that $x_i$ is included in $\mathbf{s}$*
26: $\quad\quad$ **if** $\mathbf{x}_i$ in $\mathbf{s}$ : $\Delta \ell \leftarrow \log p_i$ **else** $\Delta \ell \leftarrow \log(1 - p_i)$
27: $\quad\quad \ell \leftarrow \ell - \Delta \ell \triangleright$ *update $\ell$ with minus log-probability*
28: $\quad$ **return** $\ell$
29: **procedure** LOSSDED($\mathbf{s}, \mathbf{y}, p_{\text{ded}}$)
30: $\quad \triangleright$ *loss is negative log-prob of deduction under model*
31: $\quad \ell \leftarrow -\log p_{\text{ded}}(\mathbf{y} | \mathbf{s})$
32: $\quad \triangleright$ $\log p_{ded}(\mathbf{y} | \mathbf{s})$ *sums log-probabilities of tokens in $\mathbf{y}$*
33: $\quad$ **return** $\ell$

---

**Training.** Algorithm 7 elaborates how the SLM selection and deduction models are trained. We use prompt-learning because we do not want to distort the pretrained weights too much. It is well known that pretrained language models have already captured substantial amounts of commonsense knowledge such as hypernymy (A is a type of B) and meronymy (A is part of B) (Richardson and Sabharwal, 2020); we would like to keep such knowledge to benefit our settings.

## B.4 LLM Details

We give LLM details in this section. For selection, we use a large language model as a black box and prompt it to choose several different multi-premise selections from the given theory $\mathcal{T}$. The pseudocode is in Algorithm 8. Below is the prompt template:

```
# few-shot examples to demonstrate selection
# see Appendix C.6 for an example

⋮

# end of demonstration

Please refer to these examples, select four
pairs of indexes from the theory (e.g. 12 and
 3 / 12 and 6 / 12 and 7 / 12 and 11) that
can potentially help us answer the question,
no need to say anything else.

You must choose four pairs even if there are
no valid selections.

# theory and question/goal of interest
```

For deduction, we also use a large language model as a black box and prompt it to draw new deductions conditioned on a given selection $\mathbf{s}$. The pseudocode is in Algorithm 9. The prompt template is as follows:

```
# few-shot examples to demonstrate deduction
# see Appendix C.6 for an example

⋮

# end of demonstration

Please refer to these examples and generate
the inference.

# selection of statements of interest
```

---

**Algorithm 8** Selection Subroutine for LLM

---

**Hyperparam:** selection beam size $B_{\text{sel}}$
**Input:** current theory $\mathcal{T} = \{\mathbf{x}_1, \ldots, \mathbf{x}_{N+m}\}$ and goal $\mathbf{x}_0$; selection model $p_{\text{sel}}$
**Output:** selections with their scores $\{(u_k, \mathbf{s}_k)\}$
1: **procedure** SELECT($\mathcal{T}, \mathbf{x}_0, p_{\text{sel}}$)
2:     ▷ *has access to $B_{sel}$*
3:     prompt LLM to select $B_{\text{sel}}$ different multi-premise selections $\mathbf{s}$ from the theory $\mathcal{T}$
4:     ▷ *prompt templates are in Appendix B.4*
5:     each selection $\mathbf{s}$ is assigned a score $u = 0$
6:     construct list $\mathcal{S}$ to contain the multiple $(u, \mathbf{s})$
7:     **return** $\mathcal{S}$

---

## B.5 Details of Planning-Based Methods

Algorithm 10 illustrates the details of how we use explicit planning for selection. The method considers how each selection could affect future in $D$ steps. One-best search is applied in the roll-out process to simplify the

---

**Algorithm 9** Deduction Subroutine for LLM

---

**Hyperparam:** deduction beam size $B_{\text{ded}}$
**Input:** current selection $\mathbf{s}$ of statements; deduction model $p_{\text{ded}}$
**Output:** deductions with their scores $\{(v_k, \mathbf{y}_k)\}$
1: **procedure** DEDUCE($\mathbf{s}, p_{\text{ded}}$)
2:     ▷ *has access to $B_{ded}$*
3:     prompt LLM to draw $B_{\text{ded}}$ new deductions
4:     ▷ *prompt templates are in Appendix B.4*
5:     each deduction $\mathbf{y}$ is assigned a score $v = 0$
6:     construct list $\mathcal{Y}$ to contain the multiple $(v, \mathbf{y})$
7:     **return** $\mathcal{Y}$

---

**Algorithm 10** Planning for Selection

---

**Hyperparam:** deduction beam width $B_{\text{ded}}$; depth of planning $D$; planning scale $\alpha$
**Input:** current theory $\mathcal{T} = \{\mathbf{x}_1, \ldots, \mathbf{x}_{N+m}\}$ and goal $\mathbf{x}_0$; verification model $p_{\text{ver}}$ selection candidates at current step $\mathcal{S} = \{(u_k, \mathbf{s}_k)\}$; selection model $p_{\text{sel}}$ and deduction model $p_{\text{ded}}$
**Output:** selections with updated scores $\{(u_k, \mathbf{s}_k)\}$
1: **procedure** PLANS($\mathcal{T}, \mathbf{x}_0, \mathcal{S}, p_{\text{sel}}, p_{\text{ded}}, p_{\text{ver}}$)
2:     ▷ *has access to $B_{ded}$, $D$, $\alpha$*
3:     ▷ *init hypothetical extended theory*
4:     **for** $u_k, \mathbf{s}_k$ **in** $\mathcal{S}$ : $\tilde{\mathcal{T}}_k \leftarrow \mathcal{T}$
5:     **for** $u_k, \mathbf{s}_k$ **in** $\mathcal{S}$ :
6:         ▷ *iterate over all candidate selections*
7:         ▷ *find hypothetical next-step deduction*
8:         $\tilde{\mathbf{y}}_k \leftarrow$ ONEBESTDEDUCE($\mathbf{s}_k, p_{\text{ded}}$)
9:         ▷ *extend theory with new deduction*
10:        $\tilde{\mathcal{T}}_k \leftarrow \tilde{\mathcal{T}}_k + \{\tilde{\mathbf{y}}_k\}$
11:        ▷ *planning with roll-outs*
12:        ▷ *what's given by ROLLOUT is $\Delta u$ in section 3.2*
13:        $u_k \leftarrow u_k + \alpha$ ROLLOUT
14:     sort $\mathcal{S}$ in descending order of updated $u_k$
15:     **return** $\mathcal{S}$
16: **procedure** ROLLOUT
17:     ▷ *roll out $D$ steps of imaginary selection and deduction*
18:     ▷ *make in-place edits to $\tilde{\mathbf{s}}_k, \tilde{\mathbf{y}}_k, \tilde{\mathcal{T}}_k$*
19:     $f \leftarrow -\infty$         ▷ *init score of roll-out*
20:     **for** $d = 1$ **to** $D$ :     ▷ *step-by-step roll-out*
21:         $\tilde{\mathbf{s}}_k \leftarrow$ ONEBESTSELECT($\tilde{\mathcal{T}}_k, \mathbf{x}_0, p_{\text{sel}}$)
22:         $\tilde{\mathbf{y}}_k \leftarrow$ ONEBESTDEDUCE($\tilde{\mathbf{s}}_k, p_{\text{ded}}$)
23:         $\tilde{\mathcal{T}}_k \leftarrow \tilde{\mathcal{T}}_k + \{\tilde{\mathbf{y}}_k\}$
24:         **if** $p_{\text{ver}}(\mathbf{x}_0 \mid \tilde{\mathbf{y}}_k) > f : f \leftarrow p_{\text{ver}}(\mathbf{x}_0 \mid \tilde{\mathbf{y}}_k)$
25:     **return** $\log f$

---

**Algorithm 11** Planning for Deduction

---

**Hyperparam:** deduction beam width $B_{\text{ded}}$;
    depth of planning $D$; planning scale $\beta$

**Input:** current theory $\mathcal{T} = \{\mathbf{x}_1, \ldots, \mathbf{x}_{N+m}\}$ and goal $\mathbf{x}_0$; deduction candidates at current step $\mathcal{Y} = \{(v_k, \mathbf{y}_k)\}$; selection model $p_{\text{sel}}$ and deduction model $p_{\text{ded}}$; verification model $p_{\text{ver}}$

**Output:** deductions with updated scores $\{(v_k, \mathbf{y}_k)\}$

 1: **procedure** PLAND($\mathcal{T}$, $\mathbf{x}_0$, $\mathcal{Y}$, $p_{\text{sel}}$, $p_{\text{ded}}$, $p_{\text{ver}}$)
 2:    ▷ *has access to $B_{\text{ded}}$, $D$, $\beta$*
 3:    ▷ *init hypothetical extended theory*
 4:    **for** $v_k, \mathbf{y}_k$ **in** $\mathcal{Y}$ : $\tilde{\mathcal{T}}_k \leftarrow \mathcal{T}$
 5:    **for** $v_k, \mathbf{y}_k$ **in** $\mathcal{Y}$ :
 6:       ▷ *iterate over all candidate deductions*
 7:       $\tilde{\mathcal{T}}_k \leftarrow \tilde{\mathcal{T}}_k + \{\mathbf{y}_k\}$ ▷ *extend theory with deduction*
 8:       $v_k \leftarrow v_k + \beta$ ROLLOUT
 9:       ▷ ROLLOUT *is in Algorithm 10*
10:       ▷ *what's given by* ROLLOUT *is $\Delta v$ in section 3.2*
11:    sort $\mathcal{Y}$ in descending order of updated $v_k$
12:    **return** $\mathcal{Y}$

---

planning. Intuitively, a higher $\log f$ means that the future reasoning path conditioned on this selection is more likely to prove the goal. Similar to Algorithm 10, Algorithm 11 measures how the newly generated deduction could affect the future reasoning path in $D$ steps, and honors the deduction which improves the possibility of proving the goal in the future.

## C   Experiment Details

We present experiment details in this section.

### C.1   Data Statistics

The data statistics of Entailment Bank is shown in Table 5. In Version-I of Entailment Bank, there is one sample in the test set that has a theory with a single statement. We ignore this sample since it can not be dealt by our system in the normal way. The dataset can be downloaded from `https://allenai.org/data/entailmentbank`.

| SPLIT | # OF SAMPLES | MAX STEPS | AVG STEPS |
|-------|--------------|-----------|-----------|
| TRAIN | 1313 | 17 | 3.2 |
| DEV | 187 | 15 | 3.2 |
| TEST | 340 | 11 | 3.3 |

Table 5: Data statistics of Entailment Bank.

### C.2   Hyperparameters

For SLM experiments, we use "t5-small" in the Huggingface transformers (Wolf et al., 2020) library for the selection and deduction models. We use "deberta-v2-xlarge-mnli" for the verification model. We prompt tune these models, with a prompt length of 4 for the selection and deduction models, and a prompt length of 32 for the verification model. Note that for T5 models, the prompt is added to the beginning of both the encoder and the decoder (weight not shared). For the selection model, a layernorm is added before the sigmoid operation.

In training, we use the Adam (Kingma and Ba, 2015) optimizer with $\beta_1 = 0.9, \beta_2 = 0.999, \epsilon = 1e - 8, \lambda = 0$. We use learning rate $\gamma = 0.1$ for the T5 models, and $\gamma = 0.01$ for the verification model. We use a batch size of 16. We set a very large epoch number like 1000 and use the validation set to do early stopping. In practice, the best epoch is often within 100.

For LLM experiments, we use GPT-3.5-turbo model provided by OpenAI. We set the temperature to reduce randomness. We keep the default role "system" with the message "You are an AI assistant that speaks English."

We used a fixed random seed for all our data generation and training, so that our results can be easily reproduced with our codes.

During inference, we set $B_{\text{inf}} = B_{\text{ded}} = 5$ and retain the selections formed by 4 top-scored statements. We set $\alpha = 10$ and $\beta = 0.5$ to roughly match the scale of the beam score. We roll out 3 steps for selection and 2 steps for deduction. We set the maximum step to be $M = 20$.

We do not tune hyperparameters except the learning rate, and we only tune it in our first training of every model. We try [0.1, 0.01, 0.001, 0.0001] and choose the one that yields the best dev set performance.

Our experiments were run on 8 A6000 GPUs. Training takes about 1 hour. Time for inference is discussed in section 6.1.

### C.3   Details of FOL Translations

The classical approach of logical reasoning is to use formal logic systems. So we also evaluated the performance of a first-order-logic (FOL) system. Because the Entailment Bank dataset does not have human-annotated FOL translations for the natural language statements, we translated all the statements into FOL expressions using a T5 model trained on the corpus of (natural language, FOL) pairs collected by Levkovskyi and Li (2021), and then used a FOL engine to perform reasoning. This approach failed because the FOL translations are mostly of very poor quality. Here is a summary of the errors:

- inconsistency in variable naming. The FOL translations often use inconsistent variable naming, making it difficult to pattern-match relevant expressions.

- incorrect translations. Some FOL translations inaccurately represent the original sentences, resulting in a failure to capture the intended meaning. For example, "driving is a kind of skill" is incorrectly translated into "$\exists x.(\text{driving}(x) \& \exists y.(\text{vehicle}(y)\text{kind}(x, y)))$.

- syntax errors. Some FOL translations contain syntax errors, making them difficult to be process.

- missing or incomplete information. In several instances, the FOL translations do not capture all rel-

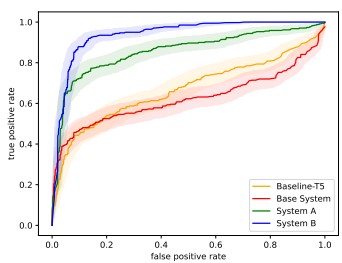
(a) ROC curves.

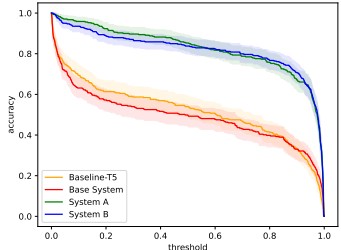
(b) Acc curves on positive examples.

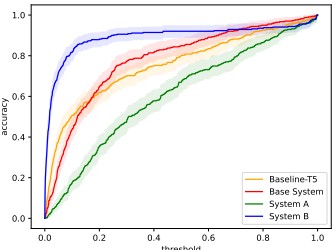
(c) Acc curves on negative examples.

Figure 8: Test results with 95% bootstrap CFs on Entailment Bank Version-I under 50% training data.

| METHOD | AUROC | AUACC$_{POS}$ | AUACC$_{NEG}$ | F1 |
|---|---|---|---|---|
| BASE SYSTEM (T5-BASE) | 0.73 (0.69, 0.77) | 0.61 (0.56, 0.65) | 0.81 (0.79, 0.84) | 0.68 (0.66, 0.71) |
| SYSTEM A (T5-BASE) | 0.91 (0.89, 0.93) | 0.90 (0.88, 0.92) | 0.62 (0.58, 0.65) | 0.85 (0.83, 0.87) |
| SYSTEM B (T5-BASE) | 0.94 (0.93, 0.96) | 0.89 (0.87, 0.91) | 0.84 (0.80, 0.87) | 0.90 (0.88, 0.91) |
| BASE SYSTEM (DENOISE) | 0.55 (0.50, 0.59) | 0.39 (0.35, 0.43) | 0.83 (0.80, 0.85) | 0.67 (0.67, 0.67) |
| SYSTEM A (DENOISE) | 0.88 (0.85, 0.90) | 0.87 (0.85, 0.89) | 0.55 (0.52, 0.59) | 0.83 (0.81, 0.85) |
| SYSTEM B (DENOISE) | 0.93 (0.91, 0.95) | 0.83 (0.80, 0.86) | 0.88 (0.85, 0.90) | 0.85 (0.84, 0.86) |

Table 6: Test results with 95% bootstrap CFs on Entailment Bank Version-I.

| METHOD | VERSION-I | VERSION-II |
|---|---|---|
| BASE SYSTEM (T5-BASE) | 0.65 (0.60, 0.70) | 0.26 (0.22, 0.31) |
| SYSTEM A (T5-BASE) | 0.88 (0.85, 0.91) | 0.45 (0.39, 0.50) |
| SYSTEM B (T5-BASE) | 0.91 (0.88, 0.94) | 0.55 (0.50, 0.60) |
| BASE SYSTEM (DENOISE) | 0.46 (0.40, 0.52) | 0.27 (0.22, 0.32) |
| SYSTEM A (DENOISE) | 0.83 (0.80, 0.87) | 0.40 (0.35, 0.46) |
| SYSTEM B (DENOISE) | 0.90 (0.87, 0.93) | 0.67 (0.62, 0.72) |
| BASE SYSTEM (VERSION-II) | - | 0.49 (0.44, 0.54) |
| SYSTEM A (VERSION-II) | - | 0.73 (0.69, 0.78) |
| SYSTEM B (VERSION-II) | - | **0.80** (0.76, 0.84) |

Table 7: Test accuracy with 95% bootstrap CFs in multiple-choice QA. A random guess gives 25% accuracy. The systems in the third block were trained on Version-II training data.

evant information from the original sentences. For example, it may leave out an entity or quantifier.

This analysis reveals a fundamental need for tools that work directly with natural language statements for reasoning like ours.

### C.4 Results of Ablation Studies

Figure 8 shows the results of the systems trained on 50% training data. Some results of ablation studies described in section 6.1 are shown in Table 6 and Table 7.

### C.5 Examples of Prompts for GPT-3

In section 6.1, we used three kinds of prompts for GPT-3: 0-shot, 5-shot and COT. In this section, we provide some examples of these prompts.

An in-context demonstration is

```
Based on the statements that:
the earth rotating on its axis causes stars /
 the moon to appear to move across the sky at
night.
```

```
diurnal motion is when objects in the sky
appear to move due to earth 's rotation on
its axis.
stars appear to move relative to the horizon
during the night.

Which of the following conclusions can be
inferred?
0. earth rotating on its axis causes horizon
of stars and night on earth.
1. earth 's horizon on its rotating axis
causes stars to occur in new york night.
2. the earth revolving around the axis causes
 stars to appear in different night in the
sky at different horizon of year.
3. the earth rotating on its axis causes
stars to appear to move relative to the
horizon during the night.

A: 3.
```

For COT prompting, we used the ground-truth reasoning path for the correct choice as the "chain-of-thought", so the last line of the (say) above example will be:

```
Reason: diurnal motion is when objects in the
```

| METHOD | DEPTH=1 | DEPTH=3 | DEPTH=5 |
|---|---|---|---|
| SYSTEM A WITH MODIFIED PROOF SCORE | **0.90** (0.84, 0.95) | 0.55 (0.45, 0.65) | **0.55** (0.45, 0.65) |
| SYSTEM A WITH ORIGINAL PROOF SCORE | 0.85 (0.78, 0.92) | **0.64** (0.54, 0.74) | 0.5 (0.41, 0.6) |
| SYSTEM B TRAINED ON ENTAILMENT BANK | 0.87 (0.80, 0.94) | 0.54 (0.44, 0.64) | 0.46 (0.36, 0.56) |

Table 8: Results of ablation studies on PrOntoQA with 95% bootstrap CFs.

```
 sky appear to move due to earth 's rotation
on its axis & stars appear to move relative
to the horizon during the night -> int1:
stars appearing to move relative to the
horizon during the night is an example of
diurnal motion; int1 & the earth rotating on
its axis causes stars / the moon to appear to
 move across the sky at night -> the earth
rotating on its axis causes stars to appear
to move relative to the horizon during the
night.

A:3.
```

## C.6 Experiment Details on PrOntoQA

The PrOntoQA data has three subsets of different "depths". The "depth" denotes the number of ground-truth reasoning steps so a "deeper" subset is harder. For each depth, we draw (using the released data generation code of Saparov and He (2023)) 5 training examples and 100 test examples.

For the experiments on PrOntoQA, our final verification is performed by a few-shot-prompted GPT-3.5: it reads the extended theory and judges whether the given goal is proved. By doing this, we do not need to tune a threshold for the proof scores given by the verification model (although those scores are still very important in the process of explicit planning). In this dataset, the non-provable goals are often definitively disapprovable. So we would like the explicit planning to favor not only the future steps that have large proof scores but also those of large *contradiction* scores. Therefore, we replace the proof score $f$ in the planning procedure by the generalized score $g$ defined below

$$g(\mathcal{T}, \mathbf{x}_0) \stackrel{\text{def}}{=} \max_n \max(p_{\text{ver}}(\mathbf{x}_0 \mid \mathbf{x}_n), p_{\text{con}}(\mathbf{x}_0 \mid \mathbf{x}_n)) \tag{4}$$

where $p_{\text{con}}(\mathbf{x}_0 \mid \mathbf{x}_n)$ is the probability of "$\mathbf{x}_n$ contradicts $\mathbf{x}_0$" given by the pretrained DeBERTa. Table 8 shows how much this modification helps: if we do not use $g$, the average performance doesn't change but we will suffer a higher variance. Table 8 also shows the results of using System B trained on Entailment Bank data. We did this to see if the verification model could generalize to out-of-domain data. In this experiment, it hurts the performance.

In this section, we also show the prompts for GPT-3.5 used in the experiments in section 6.3. For selection and deduction, we employed 5-shot prompting to enhance the model's comprehension. An in-context training example for 5-shot selection prompt is

```
Contexts:
0.Every tumpus is not earthy.
1.Wumpuses are not red.
2.Wumpuses are vumpuses.
3.Each vumpus is bitter.
4.Vumpuses are zumpuses.
5.Every zumpus is cold.
6.Zumpuses are numpuses.
7.Numpuses are aggressive.
8.Numpuses are dumpuses.
9.Dumpuses are opaque.
10.Dumpuses are yumpuses.
11.Yumpuses are not small.
12.Each yumpus is a rompus.
13.Every rompus is earthy.
14.Each rompus is a jompus.
15.Jompuses are metallic.
16.Each jompus is an impus.
17.Alex is a dumpus.
Question:True or false: Alex is not earthy.

Selection:17 and 10 / 17 and 1 / 17 and 13 /
17 and 15.
Alex is a dumpus and dumpuses are yumpuses.
Alex is a dumpus and wumpuses are not red.
Alex is a dumpus and every rompus is earthy.
Alex is a dumpus and jompuses are metallic.
```

An in-context example for deduction prompt is

```
We know that: Sally is a tumpus and each
tumpus is hot.
Inference: Sally is hot.
```

Note that we didn't let GPT to propose multiple deductions in this experiment because the no-planning deduction is almost always correct as long as the selection is correct.