# OpenReview forum: "Explicit Planning Helps Language Models in Logical Reasoning"
_EMNLP/2023/Conference — EMNLP 2023 Main_

### Official Review · Reviewer_661C · 2023-07-28

**Typos Grammar Style And Presentation Improvements:** 1. Analysis-IV
**Soundness:** 4

**Excitement:**

4: Strong: This paper deepens the understanding of some phenomenon or lowers the barriers to an existing research direction.

**Missing References:**

[1] Peter Clark, Oyvind Tafjord, and Kyle Richardson. 2021. Transformers as soft reasoners over language. In Proceedings of the Twenty-Ninth International Joint Conference on Artificial Intelligence (IJCAI'20). Article 537, 3882–3890.

[2] Gabriele Picco, Thanh Lam Hoang, Marco Luca Sbodio, and Vanessa Lopez. 2021. Neural Unification for Logic Reasoning over Natural Language. In Findings of the Association for Computational Linguistics: EMNLP 2021, pages 3939–3950, Punta Cana, Dominican Republic. Association for Computational Linguistics.

**Paper Topic And Main Contributions:**

This paper combines language model and explicit planning for multiple-step logical reasoning. For logical reasoning, planning offers two benefits: lowering the number of reasoning steps and interpretability. There is also a challenge of planning in models: model exploitation, which causes the model to be overconfident about some wrong reasoning paths. This paper presents a framework for logical reasoning that combines language models and planning. The framework is based on a base model and two improvements. The base model consists of models for deduction using beam search; the first improvement introduces planning by modifying scores in the base model; the second improvement adds a contrastive loss and KL-divergence to the original loss function. The method applies to both small models like T5 and LLMs like GPT-3.5. Experiments and analysis shows that planning does help deduction.

**Questions For The Authors:**

A. In analysis-II, what is the data that supports the claim that “without Ω, System-B 602 performs worse than System-A”?

B. Figure 1:Can you explain why each deduction step (e.g., x2x3->x5) is valid? It seems none of these are actually deductions.

C. Compared to baseline methods, how much more computation does this method cost?

**Reasons To Accept:**

1. The idea of combining LM and planning is novel and meaningful.
2. The idea is supported by detailed description of the method and extensive experiments and analysis.
3. The method applies to both small models and large models, which is a practical advantage (although the cost may be higher than other methods).

**Reasons To Reject:**

1. This paper is not well-organized. Much important information is presented in the appendix, like figures of models (fig. 6) and the results of analysis (fig. 8, 9). There also seems to be two sections for related work (sec 3.4 and 5).
2. The baselines methods are limited. For small models, the method is based on tuning, so comparing to prompted GPT-3.5 is not convincing. Instead, the paper may include some tuning baselines for deductive reasoning like RuleTaker [1] and Neural Unification [2].

**Reproducibility:**

4: Could mostly reproduce the results, but there may be some variation because of sample variance or minor variations in their interpretation of the protocol or method.

**Reviewer Confidence:**

3: Pretty sure, but there's a chance I missed something. Although I have a good feel for this area in general, I did not carefully check the paper's details, e.g., the math, experimental design, or novelty.

---

> ### Author Rebuttal · Authors · 2023-08-28
>
> Thank you for your detailed and constructive feedback! Your comments are valuable for us to improve the paper.
>
> The submitted 8-page version was rather compressed. After submission, we continued to revise it for clarity and completeness, making full use of the extra page in the main text that is allowed by camera-ready. In camera-ready, we will take your suggestions as much as possible, bringing important materials back to the main text.
>
> **About Reasons To Reject**
>
> > The baselines methods are limited. For small models, the method is based on tuning, so comparing to prompted GPT-3.5 is not convincing. Instead, the paper may include some tuning baselines for deductive reasoning like RuleTaker [1] and Neural Unification [2].
> > [1] Peter Clark, Oyvind Tafjord, and Kyle Richardson. 2021. Transformers as soft reasoners over language. In Proceedings of the Twenty-Ninth International Joint Conference on Artificial Intelligence (IJCAI'20). Article 537, 3882–3890.
> > [2] Gabriele Picco, Thanh Lam Hoang, Marco Luca Sbodio, and Vanessa Lopez. 2021. Neural Unification for Logic Reasoning over Natural Language. In Findings of the Association for Computational Linguistics: EMNLP 2021, pages 3939–3950, Punta Cana, Dominican Republic. Association for Computational Linguistics.
>
> We ran new experiments for you! We will add them into camera-ready.
>
> RuleTaker trains a RoBerta to perform logical reasoning as binary classification (provable or not), without spelling out the reasoning process. On Entailment Bank Version-I:
> - it gets 0.84 F1 on the binary classification task;
> - it gets 0.83 accuracy on the multiple-choice QA task.
>
> This performance is better than System-A but worse than System-B. (Though System-A provides an interpretable reasoning path.)
>
> We also evaluated this new baseline on QASC (like Ghosal et al. EMNLP 2022, whose setting is moderately different). QASC only requires one-step reasoning, so this discriminative baseline gets 0.96 accuracy, better than all our systems. This is consistent with findings of Clark et al. 2020 and Zhang et al. 2023: Transformers work well as soft reasoners for shallow reasoning, but tend to suffer for deep reasoning. However, this is orthogonal to our main claim (i.e., explicit planning helps).
>
> We have also experimented with Neural Unification (Picco et al. EMNLP 2021). But this experiment can not be finished by August 29. We expect to finish it by the end of the author-reviewer discussion period, and will post the results via Official Comments as soon as possible.
>
> References:
>
> Ghosal et al. EMNLP 2022, https://arxiv.org/abs/2210.16495
>
> Zhang et al. IJCAI 2023, https://arxiv.org/abs/2205.11502
>
> > This paper is not well-organized. Much important information is presented in the appendix, like figures of models (fig. 6) and the results of analysis (fig. 8, 9).
>
> We will reorganize the paper for camera-ready.
>
> Due to the page limit, we had to leave some important materials to appendices. With the extra page allowed by camera-ready, we can bring them back to the main paper.
>
> > There also seems to be two sections for related work (sec 3.4 and 5).
>
> We will reorganize Section 3.4:
> - some remarks will go to Introduction and Section 3.1 to 3.3;
> - related-work-style content will go to Related Work (Section 5).
>
> **About Typos Grammar Style And Presentation Improvements**
>
> > Analysis-IV: Since this analysis does not produce results, it could be abbreviated as a footnote or moved to appendix.
>
> Great idea!
> We will move it to Appendix C.5, making room for more important materials.
>
> > Some sentences may be informal and redundant, like line 266-267 and line 292-293.
>
> We will delete these, and fix other such cases.
>
> **About Questions For The Authors**
>
> > In analysis-II, what is the data that supports the claim that “without Ω, System-B performs worse than System-A”?
>
> We accidentally deleted it when shortening the main paper. Sorry! We will add it back for camera-ready. On Entailment Bank Version-I, the results are
> | | $\text{AUROC}$ | $\text{AUACC}_{\text{pos}}$ | $\text{AUACC}_{\text{neg}}$ |
> |-|-|-|-|
> | System-A | 0.88 | 0.86  | 0.54 |
> | System-B without $\Omega$ | 0.79 | 0.68  | 0.65 |
> | System-B | 0.96  | 0.87 | 0.89 |
>
> > Figure 1: Can you explain why each deduction step (e.g., x2x3->x5) is valid? It seems none of these are actually deductions.
>
> Sure! We'd love to.
>
> This example is chosen from Entailment Bank (Clark et al. 2021), in which each deduction step is human annotated.
>
> Our way of thinking of these deductions is to think of their formal logic representations.
>
> For example, one may represent x2 and x3 in Datalog format as:
> - eat(eagle, rabbit)   % Eagles eat rabbits.
> - be(rabbit, animal)  % Rabbits are animals
>
> Suppose there is a rule:
> - eat(X, Y) :- eat(X, Z), be(Z, Y) % If X eats Z and Z is a kind of Y, then X eats Y
>
> Then we can deduce:
> - eat(eagle, animal)  % Eagles eat animals.
>
> Has the absence of the rules confused you? Entailment Bank assumes a "common sense" world and does not explicitly provide rules.
>
> Or, are you confused by its quantifier? Entailment Bank uses casual natural languages, in which quantifiers are often omitted. For example, "Eagles eat animals." means "There are animals that eagles eat.", but not "Eagles eat all animals."
>
> These characteristics make Entailment Bank less formal, less rigorous, but more colloquial and human-like (compared to formal logic statements and synthetic data like RuleTaker). In some sense, Entailment Bank exhibits a better match with how humans do logical reasoning in everyday conversations, in which information is often incomplete and languages are often casual and imprecise. This is why we use it as our primary testbed and example in Figure 1.
>
> We will clarify it in camera-ready.
>
> Note that we also experimented with other kinds of datasets. The PrOntoQA dataset (which is similar to the data of RuleTaker) doesn't make any "world assumption", and each of its theories includes rules. Their rules may only hold in fictional worlds; their deductions are more formal and more rigorous.
>
> > Compared to baseline methods, how much more computation does this method cost?
>
> Analogous to System-I vs. System-II thinking, our method is more deliberative but slower.
> In theory, at each step of proving a goal,
> - no-planning base system needs $3 B$ operations (select, deduce, verify), where $B$ is the buffer size of beam search;
> - our planning-based system needs $3 B + 3 B E D + 3 B E D$ operations: for each candidate in buffer, we need to examine its E possible expansions (selection or deduction), and roll out D steps of future reasoning (with beam size = 1) for each expansion;
> - as a result, planning brings a performance enhancement at $1 + 2 E D$ times of cost.
>
> Our implementation (which we will release for camera-ready) is efficient due to careful tensorization and parallelism. In our experiments, we used $E=5$ and $D=2$, which should suffer $1 + 2 E D = 21$ times of slowdown. But the actual average wall-clock time of proving a goal is:
> - 2.8 seconds for the base system;
> - 31 seconds for the planning system, which is only 11 times slow.
>
> We were honest about this efficiency limitation; please see Limitations and Appendix.A of our paper. While carrying out this research project, we have been following the general principle of "make it work, make it right, make it fast" (which seems to be followed by many lines of great research in our field):
> - [make it work] we develop an intuitive and effective way to integrate explicit planning into multiple-step logical reasoning;
> - [make it right] we carefully design a training strategy to protect planning from being misguided by imperfect verification model;
> - [make it fast] it will be non-trivial work to make it fast, so we leave it as future work. However, we have proposed a few promising avenues in Appendix.A Future Extensions.

---

### Official Review · Reviewer_Mxjp · 2023-08-06

**Soundness:** 4

**Excitement:**

4: Strong: This paper deepens the understanding of some phenomenon or lowers the barriers to an existing research direction.

**Paper Topic And Main Contributions:**

This paper proposes a method that perform multi-step logical reasoning by employing explicit planning into its inference procedure. The system significantly outperforms other competing methods on EntailmentBank and QASC. The authors also propose a training strategy that safeguards the planning process from being led astray by spurious features. Extensive empirical studies demonstrate that explicit planning plays an important role in the system's performance.

**Questions For The Authors:**

1. the authors should consider improving figure 1.  The figure basically looks the same as the selection-inference paper. Personally, I didn't get anything out from figure 1.

**Reasons To Accept:**

1. The method makes an interesting connection between logical reasoning and planning / contrastive learning
2. The empirical results are pretty strong - small models can match the performance of GPT-3

**Reasons To Reject:**

1. The writing and presentation of the paper can be improved.  I didn't have any idea of the method until I really read through the method section.  The paper would benefit from first giving a high-level overview of the method at the beginning of the paper than just saying they do planning.
2. The method is quite complex, making people question whether this can really be deployed in the real-world. However, I still think there is value in studying this.  Not too much of a concern for me.

**Reproducibility:**

3: Could reproduce the results with some difficulty. The settings of parameters are underspecified or subjectively determined; the training/evaluation data are not widely available.

**Reviewer Confidence:**

3: Pretty sure, but there's a chance I missed something. Although I have a good feel for this area in general, I did not carefully check the paper's details, e.g., the math, experimental design, or novelty.

---

> ### Author Rebuttal · Authors · 2023-08-28
>
> Thank you for your constructive feedback and being supportive!
>
> The submitted 8-page version was rather compressed. After submission, we continued to revise it for clarity and completeness, making full use of the extra page in the main text that is allowed by camera-ready. In camera-ready, we will take your suggestions as much as possible.
>
> > The writing and presentation of the paper can be improved. I didn't have any idea of the method until I really read through the method section. The paper would benefit from first giving a high-level overview of the method at the beginning of the paper than just saying they do planning.
>
> This is a great suggestion. Thank you!
>
> Specifically, we plan to extend the first paragraph of Introduction, briefly describing the architecture of our full system and explaining how planning is integrated. Then, what follows is the paragraph of "Why planning?".
>
> > the authors should consider improving figure 1. The figure basically looks the same as the selection-inference paper. Personally, I didn't get anything out from figure 1.
>
> We will improve Figure 1.
>
> Initially, we aim to use Figure 1 to illustrate the problem and how a human may solve it. It looks like Creswell et al. 2023 (selection-inference) because that method was inspired by the human reasoning process as well. The text about "human reasoning" was deleted when we were trying to fit our paper (which has a lot of materials) into an 8-page submission. With the extra page allowed by camera-ready, we could easily add the relevant content back and make the figure more useful.
>
> In addition, we will explore ways to add ingredients into the figure, enabling it to show how our planning-based framework works. This may make it more useful. Before submission, we tried a few ways but they all looked cluttered. Then we decided to break the cluttered figure to Figure 1 (which illustrates the problem) and Figure 2 (which illustrates planning). But there should be ways to extend Figure 1 to show planning on a higher level than Figure 2.

---

### Official Review · Reviewer_u2em · 2023-08-10

**Soundness:** 4

**Excitement:**

4: Strong: This paper deepens the understanding of some phenomenon or lowers the barriers to an existing research direction.

**Paper Topic And Main Contributions:**

The paper presents a method for performing multi-step logical reasoning via planning. The method uses language models to construct and evaluate reasoning paths. The models are trained using a planning strategy that looks at future predictions and integrates the results into the score of past predictions to teach them to look ahead. Contrastive training is used to learn to score good and bad reasoning paths. The proposed method outperforms the baselines and achieves similar results to larger models.

**Questions For The Authors:**

Question A: Have you compared your method with the state-of-the-art of each dataset? with other planning strategies?

Question B: How did you choose the buffer size B for beam search? Does the number of ongoing paths has an impact on the performance?

**Reasons To Accept:**

The work presented is of high quality. The paper is easy to read and the ideas are explained in a clear way. All the necessary information to understand the method is provided.

The proposed method is particularly interesting and promising for tackling challenging planning and reasoning tasks, in particular the fact that the algorithm is model-agnostic and can be integrated with other LMs. The authors provide very extensive experiments demonstrating the quality of their method. The various ablation studies also clearly identify the proposed systems A and B as the reasons for the improved performance, highlighting precisely the contributions of each system.

**Reasons To Reject:**

The paper contains many references to the appendix as it contains important results that should be in the main paper, including a part of the conclusion. The back and forth while reading can be detrimental to comprehension. Fortunately, the information needed to understand the method and assess the claim is in the main content.

While the experiments are extensive, there is little comparison with existing methods. It would be great to include a comparison with the state-of-the-art for each dataset beyond the single GPT-3.5. Further comparison with other planning methods, such as those mentioned in the related work, might be interesting (as far as I understand, the only comparison is with the work of [Creswell et al., 2023]).

**Reproducibility:**

4: Could mostly reproduce the results, but there may be some variation because of sample variance or minor variations in their interpretation of the protocol or method.

**Reviewer Confidence:**

4: Quite sure. I tried to check the important points carefully. It's unlikely, though conceivable, that I missed something that should affect my ratings.

---

> ### Author Rebuttal · Authors · 2023-08-28
>
> Thank you for your constructive feedback and being supportive!
>
> > The paper contains many references to the appendix as it contains important results that should be in the main paper, including a part of the conclusion. The back and forth while reading can be detrimental to comprehension. Fortunately, the information needed to understand the method and assess the claim is in the main content.
>
> We agree with you!
>
> The submitted 8-page version was rather compressed. After submission, we continued to revise it for clarity and completeness, making full use of the extra page in the main text that is allowed by camera-ready. In camera-ready, we will take your suggestions as much as possible, bringing important materials back to the main text.
>
> > It would be great to include a comparison with the state-of-the-art for each dataset beyond the single GPT-3.5.
> > Have you compared your method with the state-of-the-art of each dataset?
>
> Most state-of-the-art methods on each dataset are not directly comparable. However, per your request, we ran new experiments, comparing against several methods that seem to be close to what may be in your mind.
>
> [New Results]
>
> The first is RuleTaker: Clark et al. IJCAI 2020. It is a discriminative method, training a RoBerta to perform logical reasoning as binary classification (provable or not), without spelling out the reasoning process. On Entailment Bank Version-I:
> - it gets 0.84 F1 on the binary classification task;
> - it gets 0.83 accuracy on the multiple-choice QA task.
>
> This performance is better than System-A but worse than System-B. (Though System-A provides an interpretable reasoning path.)
>
> We also evaluated this new baseline on QASC (like Ghosal et al. EMNLP 2022, whose setting is moderately different). QASC only requires one-step reasoning, so this discriminative baseline gets 0.96 accuracy, better than all our systems. This is consistent with findings of Clark et al. 2020 and Zhang et al. 2023: Transformers work well as soft reasoners for shallow reasoning, but tend to suffer for deep reasoning. However, this is orthogonal to our main claim (i.e., explicit planning helps).
>
> We have also experimented with Neural Unification (Picco et al. EMNLP 2021). But this experiment can not be finished by August 29. We expect to finish it by the end of the author-reviewer discussion period, and will post the results via Official Comments as soon as possible.
>
> References:
>
> Clark et al. IJCAI 2020, https://arxiv.org/abs/2002.05867
>
> Ghosal et al. EMNLP 2022, https://arxiv.org/abs/2210.16495
>
> Zhang et al. IJCAI 2023, https://arxiv.org/abs/2205.11502
>
> Picco et al. EMNLP 2021, https://arxiv.org/abs/2109.08460
>
> [About State-of-the-Art]
>
> PrOntoQA is a new dataset designed to challenge LLMs, and GPT-3.5 with CoT is a very strong baseline, if not state-of-the-art, on this data.
>
> The original problem settings of Entailment Bank and QASC are different from ours, and thus the state-of-the-art methods on them are not directly applicable.
>
> The original Entailment Bank only has positive <theory, goal> pairs (i.e., goal is always provable given theory), because it was proposed to train and evaluate methods that generate proofs for provable goals. In our problem setting, one has to determine whether a goal is provable, which is out of the scope of state-of-the-art methods on this data.
>
> QASC was initially to evaluate whether one could retrieve most relevant premises for a given goal as well as whether one could determine the truth value of a goal given the retrieval. The second subtask is similar to our problem, but the state-of-the-art methods on QASC all focus on optimizing for the first subtask, which is more of a bottleneck for the overall performance.
>
> > Further comparison with other planning methods, such as those mentioned in the related work, might be interesting (as far as I understand, the only comparison is with the work of [Creswell et al., 2023]).
> > with other planning strategies?
>
> We are sorry that our current presentation has misled you: none of the methods mentioned in Related Work performs planning.
> Bostrom et al. 2022 is similar to the SLM version of our no-planning base system.
> Creswell et al. 2023 is similar to the LLM version of our no-planning base system.
> We will clarify it in the camera-ready.
>
> > How did you choose the buffer size B for beam search? Does the number of ongoing paths have an impact on the performance?
>
> A beam size of 5 is a common choice in text generation, so we adopted this convention and didn't tune it at all.
> Per your request, we tried B = 2, 3, 10 on Entailment Bank and found that:
> - a smaller B tends to decrease the accuracy on positive samples, but increase it on negative samples.
> - a larger B tends to increase the accuracy on positive samples, but decreases it on negative samples.
> - either way, the accuracy changes are small.
> - overall, we only see tiny changes in AUROC (which depends on accuracy on both positive and negative samples).

---

### Meta-Review · Area_Chair_TeEm · 2023-09-14

**Recommendation:** 5

**Metareview:**

This paper presents a new method combine the strengths or large models with explicit planning. A contrastively trained model scores reasoning paths to prevent spurious correlations. Reviewers assessed the work as "high quality", with "pretty strong empirical results" and "novel and meaningful". They also raised concerns about presentation and clarity, as well as missing comparison with SOTA baselines, which were addressed in the rebuttal.

---

### Decision · Program_Chairs · 2023-10-07

**Decision:**

Accept-Main

**Comment:**

This paper presents a new method combine the strengths or large models with explicit planning. A contrastively trained model scores reasoning paths to prevent spurious correlations. Reviewers assessed the work as "high quality", with "pretty strong empirical results" and "novel and meaningful". They also raised concerns about presentation and clarity, as well as missing comparison with SOTA baselines, which were addressed in the rebuttal.